# Learning Transformer Programs

**Dan Friedman    Alexander Wettig    Danqi Chen**
Department of Computer Science & Princeton Language and Intelligence
Princeton University
`{dfriedman,awettig,danqic}@cs.princeton.edu`

## Abstract

Recent research in mechanistic interpretability has attempted to reverse-engineer Transformer models by carefully inspecting network weights and activations. However, these approaches require considerable manual effort and still fall short of providing complete, faithful descriptions of the underlying algorithms. In this work, we introduce a procedure for training Transformers that are mechanistically interpretable by design. We build on RASP [Weiss et al., 2021], a programming language that can be compiled into Transformer weights. Instead of compiling human-written programs into Transformers, we design a modified Transformer that can be trained using gradient-based optimization and then automatically converted into a discrete, human-readable program. We refer to these models as *Transformer Programs*. To validate our approach, we learn Transformer Programs for a variety of problems, including an in-context learning task, a suite of algorithmic problems (e.g. sorting, recognizing Dyck-languages), and NLP tasks including named entity recognition and text classification. The Transformer Programs can automatically find reasonable solutions, performing on par with standard Transformers of comparable size; and, more importantly, they are easy to interpret. To demonstrate these advantages, we convert Transformers into Python programs and use off-the-shelf code analysis tools to debug model errors and identify the "circuits" used to solve different sub-problems. We hope that Transformer Programs open a new path toward the goal of intrinsically interpretable machine learning.[1]

## 1   Introduction

Transformers [Vaswani et al., 2017] have become the predominant neural network architecture in machine learning, representing the state-of-the-art in natural language processing (NLP) and increasingly in computer vision, molecular modeling and other domains. Prominently, large Transformer-based language models [LLMs; Brown et al., 2020] have demonstrated impressive general-purpose capabilities and are now widely deployed as components in user-facing AI applications such as chatbots, search engines, and code assistants. However, these systems are fundamentally limited by a lack of interpretability, which makes them difficult to audit, debug, and maintain. This black-box quality poses practical challenges and limits the usefulness of these models in high-stakes applications.

As a result, a considerable body of work has aimed at improving our understanding of Transformers. In NLP, much of this work has focused on pre-trained Transformer language models such as BERT [Devlin et al., 2019], using a variety of post-hoc methods, including analyzing attention patterns [Clark et al., 2019] and probing hidden representations [e.g., Tenney et al., 2019, Belinkov, 2022]. These post-hoc approaches can provide partial insight into model behavior but have also been shown to be misleading [Jain and Wallace, 2019, Bolukbasi et al., 2021]; in any case, they do not provide a complete or faithful description of the algorithm that the model implements. More recently,

---

[1]Our code is available at `https://github.com/princeton-nlp/TransformerPrograms`, along with a number of example Transformer Programs.

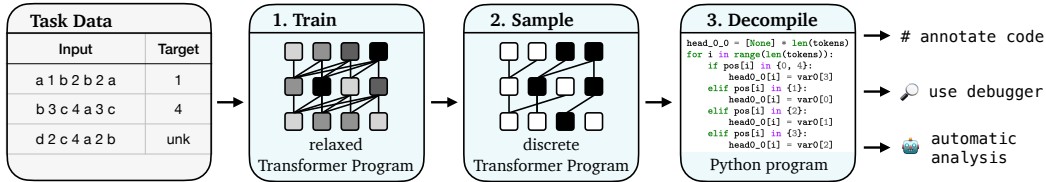

Figure 1: We design a modified Transformer that can be trained on data and then automatically discretized and converted into a human-readable program. The program is functionally identical to the Transformer, but easier to understand—for example, using an off-the-shelf Python debugger.

research on mechanistic interpretability has attempted to gain an algorithmic understanding of these models, with the goal of reverse-engineering Transformers into human-interpretable components. This line of work includes efforts to characterize "circuits" of attention heads [Elhage et al., 2021, Olsson et al., 2022, Wang et al., 2023, Nanda et al., 2023]; align network representations with symbolic causal models [Geiger et al., 2023]; interpret feed-forward layers [Nostalgebraist, 2020, Geva et al., 2022]; and localize and edit factual information [Meng et al., 2022]. However, such methods require extensive manual effort and can be impeded by the inherent complexity of the underlying models [e.g. McGrath et al., 2023].

In this work, instead of attempting to explain black-box models, we aim to train Transformers that are mechanistically interpretable by design. We take inspiration from RASP [Weiss et al., 2021], a programming language for characterizing Transformer operations, and Tracr [Lindner et al., 2023], a compiler for converting RASP programs into Transformer networks. RASP provides a conceptual framework for mapping between Transformer components and human-readable code. However, whereas prior work has compiled human-written programs into Transformers, our goal is to train Transformers using gradient-based optimization and then automatically decompile them into human-readable programs.

To this end, we propose a method for learning *Transformer Programs*—Transformers that are constrained to implement human-interpretable algorithms. First, we introduce a set of constraints that restrict the Transformer to lie within an interpretable subspace of the parameter space, by which we mean the subset of parameter values that can be mapped to a family of human-readable programs. Second, we develop a continuous relaxation scheme for learning these programs. Transformer Programs can be deterministically mapped to a program in a human-readable programming language, like Python. The Python program is functionally identical to the Transformer, but considerably easier to interpret—for example, we can debug errors by setting breakpoints using the standard Python debugger. Our overall approach is illustrated in Figure 1.

We validate our approach by learning Transformer Programs for a variety of problems, including an in-context learning task; the set of algorithmic problems introduced by Weiss et al. [2021]; and NLP benchmarks for named entity recognition and text classification. While discrete optimization introduces some challenges, we show that the Transformer Programs achieve reasonable performance relative to standard Transformers of comparable size, and, unlike standard Transformers, they are easy to interpret. We illustrate this point by converting Transformers into Python programs and using off-the-shelf code analysis tools to extract the "circuits" used to solve different sub-problems, and to debug model errors. Overall, we aim to demonstrate that Transformer Programs represent a promising first step towards building intrinsically interpretable machine learning models.

## 2 Background

**Transformer architecture.** The Transformer [Vaswani et al., 2017] is a neural network architecture for processing sequential data. The input to the Transformer is a sequence of tokens $\boldsymbol{w} = w_1, \ldots, w_N \in \mathcal{V}$ in a discrete vocabulary $\mathcal{V}$. At the input layer, the model maps the tokens to a sequence of $d$-dimensional embeddings $\mathbf{x}_0 \in \mathbb{R}^{N \times d}$. In the standard Transformer, this initial embedding is defined as the sum of a learned token embedding and a positional embedding. Each subsequent layer $i$ consists of a multi-head attention layer (MHA) followed by a multilayer perceptron layer (MLP): $\mathbf{x}_i = \mathbf{x}_{i-1} + \mathrm{MLP}_i(\mathbf{x}_{i-1} + \mathrm{MHA}_i(\mathbf{x}_{i-1}))$.[2] Following the presentation

---

[2]The standard Transformer also includes a layer-normalization layer [Ba et al., 2016], which we omit here.

of Elhage et al. [2021], multi-head attention can be written as

$$\text{MHA}(\mathbf{x}) = \sum_{h=1}^{H} \text{softmax}(\mathbf{x}\mathbf{W}_Q^h(\mathbf{x}\mathbf{W}_K^h)^\top)\mathbf{x}\mathbf{W}_V^h\mathbf{W}_O^h,$$

where $H$ is the number of heads, $d_h$ is the attention embedding dimension, $\mathbf{W}_Q^h, \mathbf{W}_K^h, \mathbf{W}_V^h \in \mathbb{R}^{d \times d_h}$ are referred to as the *query*, *key*, and *value* projections respectively, and $\mathbf{W}_O^h \in \mathbb{R}^{d_h \times d}$ projects the output value back to the model dimension. By default, we assume that attention is bi-directional, corresponding to a Transformer encoder, but we also support causal masking. The MLP layer operates at each position independently; in the standard Transformer, it is a two-layer feedforward network: $\text{MLP}(\mathbf{x}) = \sigma(\mathbf{x}\mathbf{W}_1)\mathbf{W}_2$, where $\mathbf{W}_1 \in \mathbb{R}^{d \times d_m}, \mathbf{W}_2 \in \mathbb{R}^{d_m \times d}$, and $\sigma$ is a non-linear activation, commonly the ReLU function. The output of the model is a sequence of token embeddings, $\mathbf{x}_L \in \mathbb{R}^{N \times d}$. The model is typically trained end-to-end on a prediction task by applying a linear classifier to the final-layer token embeddings.

**Transformer circuits.** Transformer circuits [Elhage et al., 2021] are an abstraction for characterizing how a neural network processes information. Informally, a Transformer can be viewed as a series of nodes (attention heads and feed-forward layers) reading and writing information to the *residual stream*, which refers to the sequence of token embeddings $\mathbf{x}_0, \ldots, \mathbf{x}_L \in \mathbb{R}^{N \times d}$ computed at each layer. Each node "reads" from the residual stream via a projection matrix $\mathbf{W}_{\text{in}} \in \mathbb{R}^{d \times d_h}$; calculates some function $f(\mathbf{x}_{i-1}\mathbf{W}_{\text{in}})$; and then "writes" to the residual stream via another projection: $\mathbf{x}_i = \mathbf{x}_{i-1} + f(\mathbf{x}_{i-1}\mathbf{W}_{\text{in}})\mathbf{W}_{\text{out}}$, with $\mathbf{W}_{\text{out}} \in \mathbb{R}^{d_h \times d}$. Mechanistic interpretability seeks to find the circuits within a Transformer that implement individual behaviors, but this goal is often difficult in practice; for example, different attention heads may write to the same subspace of the residual stream, making it difficult to disentangle how information flows from one component to another.

**Programming languages for Transformer.** RASP [Weiss et al., 2021] is a programming language consisting of a set of function primitives for defining operations on sequences. These are designed to ensure that any program written in RASP can be implemented by a Transformer network, by mapping between RASP primitives and Transformer components. Tracr [Lindner et al., 2023] is a compiler for RASP that implements this mapping in practice. The central operation is `select`, which corresponds to attention in the Transformer. `select` takes as input a sequence of `keys` $\in \mathcal{K}^N$; a sequence of `queries` $\in \mathcal{Q}^M$; and a boolean `predicate` $\in \mathcal{Q} \times \mathcal{K} \to \{0, 1\}$. The output is an attention matrix $\mathbf{A} \in \{0, 1\}^{M \times N}$ with $\mathbf{A}_{i,j} = \texttt{predicate}(\texttt{queries}_i, \texttt{keys}_j)$. The `select` operator is combined with an `aggregate` operation, which takes as input an attention matrix $\mathbf{A}$ and a sequence of `values` $\in \mathcal{V}^N$ and outputs, at each row $i$, the weighted average of `values`, with weights given by $\mathbf{A}_i$. RASP also supports arbitrary element-wise operations, which correspond to the feed-forward layers in a Transformer. RASP and Tracr provide a framework for mapping from human-interpretable programs to Transformers, but there is no clear way to map an arbitrary Transformer to a RASP program.

## 3 Learning Transformer Programs

Our main idea is to define constraints on the Transformer weights that guarantee that there is a simple, deterministic mapping between Transformer components and RASP-like programming primitives. We refer to the set of Transformers satisfying these constraints as *Transformer Programs*. To provide an overall picture our approach, we illustrate our method with a minimal Transformer Program, containing only categorical attention layers. This model was trained to perform a simple in-context learning task (described in more detail in Section 4.1) and is depicted in Figures 2 and 3. In Section 3.3, we show we extend our framework to support a broader set of operations.

### 3.1 Disentangled residual stream

Our first constraint is to ensure that the model maintains a disentangled residual stream. This means that the token embeddings encode the values of a fixed set of named variables, with each variable encoded in an orthogonal subspace. This constraint is illustrated in Figure 2. In this example, the initial input embedding encodes the values of two categorical variables, corresponding to the token and position embeddings. Each attention layer then reads a fixed set of variables from the residual stream and writes a new variable to a dedicated address.

**Reading from the residual stream.** To ensure that each module reads a fixed set of variables, we parameterize each projection matrix by a one-hot indicator vector over the available variables.

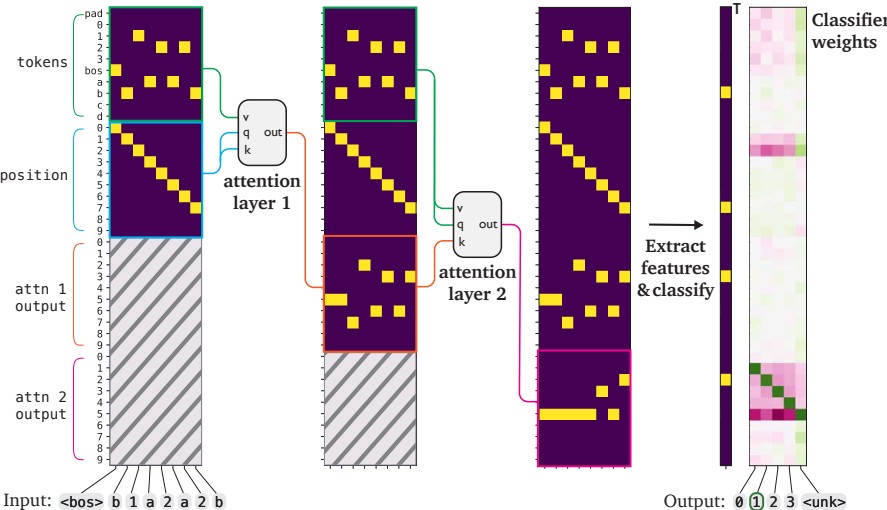

Figure 2: We constrain the Transformer to have a *disentangled residual stream*: the token embeddings encode a fixed set of discrete variables in orthogonal subspaces, and each module reads a fixed set of variables and writes a new variable to a dedicated address. Here, we depict the residual stream of a Tranformer Program that was trained on a simple in-context learning task (Section 4.1).

Suppose the residual stream encodes $m$ categorical variables, all with cardinality $k$, resulting in input embeddings $\mathbf{x} \in \{0,1\}^{N \times mk}$. Then each projection matrix $\mathbf{W} \in \mathbb{R}^{mk \times k}$ is defined by an indicator $\boldsymbol{\pi} \in \{0,1\}^m : \sum_{i=1}^m \pi_i = 1$. That is, $\mathbf{W} = [\pi_1 \mathbf{I}_k; \ldots; \pi_m \mathbf{I}_k]^\top$, where $\mathbf{I}_k$ is the $k \times k$ identity matrix. Concretely, this means that each attention head is associated with three gate vectors, $\boldsymbol{\pi}_K, \boldsymbol{\pi}_Q, \boldsymbol{\pi}_V$, defining the key, query, and value variables, respectively.

**Writing to the residual stream.** To ensure that the residual stream remains disentangled, we constrain each module to write its output to a dedicated, orthogonal subpace, taking inspiration from Tracr. In practice, we accomplish this simply by concatenating the module output to the input—i.e., if layer $i$ consists of a single attention head $h$, then $\mathbf{x}_i = [\mathbf{x}_{i-1}; h(\mathbf{x}_{i-1})]$. (This is equivalent to adding the output to the input after padding with zeros.) The output of each attention head is another categorical variable with cardinality $k$, so the final embedding dimension is then $(2 + L \times H) \times k$, where $L$ is the number of layers and $H$ is the number of heads per layer.

### 3.2 Transformer Program modules

Next, we constrain each module to implement an interpretable, rule-based mapping between inputs and outputs. Here, we describe the primary modules in our programs, categorical attention heads; we extend this framework to additional modules in Section 3.3. Categorical attention heads can be decomposed into two operations, corresponding to the select and aggregate operations in RASP: first determining which queries attend to which keys, and then aggregating the corresponding values.

**Computing the attention pattern.** First, each attention head reads one key and one query variable, and then determines which queries should attend to which keys. In RASP, this operation is defined via a boolean predicate, which maps every combination of key and query to a value in $\{0,1\}$. We implement a simple learnable predicate by associating each attention head with a one-to-one mapping between query values and key values (Figure 3). Assuming that all variables have cardinality $k$, each attention head is associated with a binary predicate matrix, $\mathbf{W}_{\text{predicate}} \in \{0,1\}^{k \times k}$, with the constraint that each row $\mathbf{W}_{\text{predicate},i}$ sums to one. The self-attention pattern is then defined by a score matrix $\mathbf{S} \in \{0,1\}^{N \times N}$, with $\mathbf{S} = \mathbf{x} \mathbf{W}_Q \mathbf{W}_{\text{predicate}} (\mathbf{x} \mathbf{W}_K)^\top$.

**Aggregating categorical variables.** We adopt a constraint from Tracr and require that each query token attends to a single key token; this constraint is necessary to ensure that the output variable will also be categorical. We enforce this by using hard attention, defining the output of head $h$ as $\mathbf{A}_i = \text{One-hot}\left(\arg\max_j \mathbf{S}_{i,j}\right)$, where $\mathbf{A}_i$ denotes the $i^{th}$ row of the attention matrix. We implement hard attention so that, in the event that there is no matching key for a query, the model defaults to


<div>

**attention layer 1**

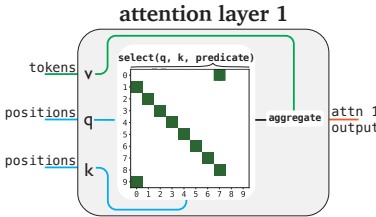

</div>
<div>

**attention layer 2**

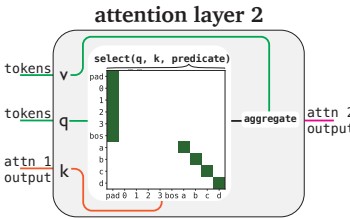

</div>
</div>

Figure 3: We further constrain each module to implement an interpretable, rule-based mapping between input and output variables. This figure depicts the categorical attention module, which defines a one-to-one correspondence between key and query variables. In this example, the first attention layer outputs the value of the `tokens` variable from the preceding position. The second attention layer uses this output to implement a simple induction head [Olsson et al., 2022], mapping each letter token to a number that is preceded by the same letter.

```python
def predicate_1(q_position, k_position):
    if q_position in {0, 8}:
        return k_position == 7
    if q_position in {1, 9}:
        return k_position == 0
    if q_position in {2}:
        return k_position == 1
    if q_position in {3}:
        return k_position == 2
    if q_position in {4}:
        return k_position == 3
    if q_position in {5}:
        return k_position == 4
    if q_position in {6}:
        return k_position == 5
    if q_position in {7}:
        return k_position == 6

attn_1_pattern = select_closest(
    positions, positions, predicate_1)
attn_1_outputs = aggregate(attn_1_pattern, tokens)
```

```python
def predicate_2(token, attn_1_output):
    if token in {
        "<pad>", "0", "1", "2", "3", ""
    }:
        return attn_1_output == "<pad>"
    if token in {"a"}:
        return attn_1_output == "a"
    if token in {"b"}:
        return attn_1_output == "b"
    if token in {"c"}:
        return attn_1_output == "c"
    if token in {"d"}:
        return attn_1_output == "d"

attn_2_pattern = select_closest(
    attn_1_outputs, tokens, predicate_2)
attn_2_outputs = aggregate(attn_2_pattern,
                           tokens)
```

Figure 4: After training, we convert the model into an equivalent Python program, modeled on RASP. This code corresponds to the attention heads illustrated in Figure 3.

attend to a beginning of sequence token; in the event that there is more than one matching key, the model attends to the closest match. More details are provided in the Appendix A.1.

**Optimization.** Consider a Transformer Program with a single categorical attention head, with $m$ input variables of cardinality $k$. This model is defined by the indicators $\boldsymbol{\pi}_K, \boldsymbol{\pi}_Q, \boldsymbol{\pi}_V$ and the predicate matrix $\mathbf{W}_{\text{predicate}}$. To learn Transformer Programs, we optimize a distribution over these discrete program weights. For each gate $\boldsymbol{\pi}_K, \boldsymbol{\pi}_Q, \boldsymbol{\pi}_V$, we define a parameter $\boldsymbol{\phi}_K, \boldsymbol{\phi}_Q, \boldsymbol{\phi}_V \in \mathbb{R}^m$. For row in the predicate matrix $\mathbf{W}_{\text{predicate}}$, we define parameters $\boldsymbol{\psi}_1, \dots, \boldsymbol{\psi}_k \in \mathbb{R}^k$, with $\boldsymbol{\psi}_i$ defining a distribution over the $i^{th}$ row. Referring to these parameters collectively as $\boldsymbol{\Phi}$, we define a distribution over discrete Transformer weights $p(\boldsymbol{\theta} \mid \boldsymbol{\Phi})$ as the product of categorical distributions, which we optimize using the Gumbel reparameterization [Jang et al., 2017]. Given a classification dataset $\mathcal{D}$ consisting of inputs $\boldsymbol{w}$ and labels $y$, we minimize the loss

$$\mathcal{L} = \mathbb{E}_{\boldsymbol{\theta} \sim p(\boldsymbol{\theta}|\boldsymbol{\Phi})} \left[ \mathbb{E}_{\boldsymbol{w}, y \sim \mathcal{D}} \left[ -\log p(y \mid \boldsymbol{w}; \boldsymbol{\theta}) \right] \right] \approx \frac{1}{S} \sum_{s=1}^{S} \mathbb{E}_{\boldsymbol{w}, y \sim \mathcal{D}} \left[ -\log p(y \mid \boldsymbol{w}; \boldsymbol{\theta}_s) \right],$$

where $\boldsymbol{\theta}_s$ is a sample from the product of Gumbel-Softmax distributions, with softmax temperature $\tau > 0$, and $S$ is a hyper-parameter denoting the number of samples per training step. Transformer Programs also include a discrete $\arg\max$ operations in hard attention, which we also relax using the Gumbel-Softmax. We anneal the Gumbel temperature over the course of training. As $\tau \to 0$, the samples are identical to one-hot samples of the corresponding categorical distribution.

**Extracting programs.** After training, we obtain a discrete model by taking the maximum likelihood weights, $\boldsymbol{\theta}^* = \arg\max_{\boldsymbol{\theta}} p(\boldsymbol{\theta} \mid \boldsymbol{\Phi})$, and applying the $\arg\max$ operation for hard attention. We then implement a deterministic mapping to convert the model to a Python program, inspired by RASP (Figure 4). Specifically, we convert each attention head $h$ into a `predicate` function, which maps each possible pair of query and key values to a value in $\{0, 1\}$. The attention operation is then

implemented by a library function, `select_closest`, which selects the closest key satisfying the predicate, or the first token in the sequence if there is no match.

## 3.3 Extensions

This framework can be extended to support additional modules, provided that the module can be mapped to a program and optimized effectively. We introduce three extensions here, providing more details and examples in Appendix A.2.

**Learning word embeddings.** For synthetic tasks with a small vocabulary (i.e., $|\mathcal{V}| < 100$), we use fixed, one-hot token embeddings. For larger-scale NLP tasks, we learn a factored categorical embedding, representing each token as the product of $m$ categorical variables, where $m$ is another hyperparameter. That is, each word embedding is the concatenation of $m$ $k$-dimensional one-hot embeddings. After training, these variables can be understood by inspecting the set of words taking on each value, which we illustrate in Figure 7.

**Aggregating numerical variables.** We also equip the model with a form of numerical attention, to facilitate operations like counting. We augment the input embeddings with a single scalar variable, which is fixed to be equal to one. In order to ensure that the resulting program is discrete, we implement a limited form of numerical attention, which guarantees that the output values are integers with a bounded range. This enables us to characterize downstream modules by enumerating the possible input values. This module loosely corresponds to the `selector_width` primitive from RASP. In Tracr and RASP, `selector_width` is implemented using one attention layer and one feed-forward layer. We "hard-code" it in the form of an attention-like head that reads categorical variables as key and query and a numerical variable as the value. As above, the module learns a binary predicate matrix mapping queries to keys, but aggregates values by taking a weighted sum. That is, given attention scores $\mathbf{S} \in \{0, 1\}^{M \times N}$ and value variable `var`, the output for the $i^{th}$ token is defined as $\sum_{j=1}^{N} \mathbf{S}_{i,j} \texttt{var}[j]$.

**Feed-forward layers.** Finally, we implement a simple feed-forward layer, designed to correspond to a lookup-table. Each feed-forward layer reads $\ell$ input variables, which are designated in advance to be either numerical or categorical variables, and outputs one new categorical variable. For example, assuming the residual stream encodes $m$ categorical variables with cardinality $k$, and $n$ numerical variables. The output of a categorical feed-forward layer is defined as $\mathbf{z} = \text{One-hot}(\arg\max(f_\theta(\mathbf{x}\mathbf{W}_{\text{in}}))$, where $\mathbf{W}_{\text{in}} \in \mathbb{R}^{(mk+n) \times \ell k}$ is a projection matrix constrained, as above, to read $\ell$ categorical variables, and $f : \mathbb{R}^{\ell k \to k}$ is an arbitrary transformation. (We implement $f$ as an MLP with one hidden layer). This module can be fully described as a look-up table, mapping $k^\ell$ possible inputs to $k$ outputs. An example of a learned MLP is illustrated in Figure 9.

## 4 Experiments

In this section, we learn Transformer Programs for a variety of tasks, including a simple in-context learning experiment (§4.1); the suite of algorithmic tasks introduced by Weiss et al. [2021] (§4.2); and two NLP tasks: named entity recognition and text classification (§4.3). Additional implementation details for all experiments are provided in Appendix §B, and we provide further ablations and analysis of the generated code in Appendix §C.

## 4.1 In-context learning

First, by way of illustration, we learn a Transformer Program for a toy task (Figure 5), designed to elicit a simple form of in-context learning [Brown et al., 2020]. The input to the model is a sequence of alternating letters and numbers that end with a letter. If the letter has appeared already in the sequence, the model should output the number that followed it. If the letter has not appeared before, the model should output an *unk* token. We train an attention-only Transformer

| Input: a1b2b2a | Target: 1 |
| Input: b3c4a3c | Target: 4 |
| Input: d2c4a2b | Target: ? |

Figure 5: Sample inputs and targets for a simple in-context learning task.

with two layers and one attention head per layer, with a vocabulary of four numbers and four letters, and trained on sequences of up to 10 tokens long. The token and position variables are fixed, one-hot indicators, meaning that the input embeddings encode two variables and the final embeddings encode four: `tokens`, `positions`, and one output variable for each attention head. The cardinality of each variable is $k = 10$. For this task, we use a causal attention mask.

| Dataset | Description | Example | $k$ | L | H | M | *Acc.* |
|---|---|---|---|---|---|---|---|
| Reverse | Reverse a string. | `reverse`("abbc") = "cbba" | 8 | 3 | 8 | 2 | 99.79 |
| Histogram | For each token, the number of occurrences of that letter in the sequence. | `hist`("abbc") = "1221" | 8 | 1 | 4 | 2 | 100.0 |
| Double hist. | For each token, the number of unique tokens with the same histogram value. | `hist2`("abbc") = "2112" | 8 | 3 | 4 | 2 | 98.40 |
| Sort | Sort the input in lexicographical order. | `sort`("cbab") = "abbc" | 8 | 3 | 8 | 4 | 99.83 |
| Most-Freq | The unique input tokens in order of frequency, using position to break ties. | `most_freq`("abbc") = "bac" | 8 | 3 | 8 | 4 | 75.69 |
| Dyck-1 | For each position $i$, is the input up until $i$ a valid string in Dyck-1 (T); a valid prefix (P); or invalid (F). | `dyck1`("()())") = "PTPTF" | 16 | 3 | 8 | 2 | 99.30 |
| Dyck-2 | The same as above, but in Dyck-2. | `dyck2`("({})(}") = "PPPTPF" | 16 | 3 | 4 | 4 | 99.09 |

Table 1: The RASP tasks, as introduced by Weiss et al. [2021]. We train Transformer Programs on small-scale instance of each task and report the number of layers (L), attention heads (H), and MLPs (M) used in the best-performing model. $k$ denotes the variable cardinality, which is fixed at the maximum sequence length for each task.

**Results.** The learned program achieves perfect accuracy on a held-out test set, and we can observe that the model learns to compose attention heads according to the *induction heads* pattern identified by Elhage et al. [2021]. The residual stream is illustrated in Figure 2, and the attention heads in Figure 3. The first attention head learns to read the `position` variable for both key and query. At each position, it attends to the key at the previous position and writes the value of the `tokens` variable as the value. The second attention head reads `tokens` as the query and the output of the previous head (`head_0_0_outputs`) as key. If the query is a letter, the head attempts to map it to a key that has the same letter written to `head_0_0_outputs`—that is, a number preceded by the same letter.

## 4.2 RASP tasks

Next, we test whether we can learn Transformer Programs for a wider variety of algorithmic tasks. We use the tasks introduced by Weiss et al. [2021] to illustrate the RASP language, which are listed in Table 1. We train on small-scale instances of each task, setting the maximum sequence length to 16 for Dyck-1 and Dyck-2, and setting the maximum sequence length and vocabulary size to 8 for the remaining tasks. As above, we use fixed one-hot token and position embeddings and set the variable cardinality $k$ to be equal to the maximum sequence length. For this setting, we introduce numerical attention and MLPs. We equip each model with an equal number of categorical and numerical attention heads, and categorical and numerical MLPs, fixing the number of MLP input variables to two, and perform a grid-search over the number of layers, attention heads, and MLPs per-layer.

**Results.** The results of this experiment are in Table 1. On five out of seven tasks, our method is able to find a program that gets more than 99% accuracy. The exceptions are Double-Histogram (98.4) and Most-Freq (75.69). These results show that, at least on short inputs, Transformer Programs can learn effective solutions to a variety of algorithmic tasks. On the other hand, in Appendix §C.1, we find that the results degrade for longer inputs, suggesting that the model may not learn robust solutions to all tasks. Additional, we can observe that the learned models often use more layers and attention heads than we might expect. For example, Weiss et al. [2021] present RASP solutions for Sort and Reverse requiring only two layers and one attention head, while our best models use three layers and eight attention heads (four categorical and four numerical). These observations indicate that the solutions found by our method might be quite different from human-written solutions.

**What kinds of programs does the model learn?** To get a better sense for how the model solves these tasks, we examine a program that the model learns for the Sort task. We analyze a model with two layers and four attention heads per layer, which achieves greater than 99% accuracy, and compile it into a Python program of approximately 300 lines, excluding the classifier weights. We find that the program uses a variety of heuristics for sorting these sequences. For example, Figure 6 depicts the code for one of the first-layer attention heads. This head looks for a 1 or a 4 in the sequence. If it finds a 1, it increases the likelihood of outputting 1 at positions one or two, and if it finds a 4, it increases the likelihood of outputting 4 at positions three through six. In Appendix Fig.11, we use an interactive Python debugger to find a more sophisticated circuit, composing attention heads at two layers. We provide more examples in Appendix §A.2 and analysis of the learned programs in Appendix §C.2. In general, we find that the model learns a variety of non-trivial solutions, composing

```
## First attention head:
# If q is early in the sequence, look for a 1.
# If q is late in the sequence, look for a 4.
def predicate_0_0(position, token):
    if position in {0}:
        return token == "3"
    if position in {1, 2}:
        return token == "1"
    if position in {3, 4, 5, 6}:
        return token == "4"
    if position in {7}:
        return token == "0"

attn_0_0_pattern = select_closest(tokens, positions,
                                  predicate_0_0)
attn_0_0_outputs = aggregate(attn_0_0_pattern, tokens)
```

Figure 6: *Left:* This code corresponds to an attention head in a Transformer Program that was trained to sort sequences of up to eight tokens long, with beginning- and end-of-sequence tokens. At early positions, this attention head checks if there is a 1 in the sequence, and at later positions it looks for a 4. *Right:* The classifier weights associated with this feature.

operations at different layers to compute higher-order patterns. These solutions include both brittle heuristics and sensible strategies for the different tasks.

**Comparison to standard Transformers.** In Appendix Fig. 10, we provide additional results comparing Transformer Programs with standard Transformers on RASP tasks with different lengths and vocabulary sizes. Standard Transformers out-perform Transformer Programs across most tasks.[3] Interestingly, we observe that both models show similar trends, with model performance degrading for tasks with longer sequences and larger vocabularies. Nevertheless, the Transformer Programs deteriorate more dramatically, with over a 50% accuracy gap relative to the standard Transformer in the most difficult setting. We discuss these scaling challenges in more detail in Section 6.

### 4.3 NLP tasks: named entity recognition and text classification

Now, we turn to standard NLP tasks including named entity recognition [Sang and De Meulder, 2003] and text classification [Pang and Lee, 2005, 2004, Zhang et al., 2015]. (Due to space constraints, we present text classification results in Appendix §C.4.) We first experiment with the English portion of the CoNLL-2003 Named Entity Recognition task [Sang and De Meulder, 2003]. This dataset consists of sentences from Reuters news articles (14,041 train/3,250 validation/3,453 test sentences). The data is labeled according to an IOB2 tagging scheme, with each word being assigned to one of four entity types (PER, ORG, LOC, MISC) or marked as not an entity (O). We filter the dataset to sentences with at most 32 words and use a vocabulary of the 10,000 most common words, replacing the remaining words with an *unknown* token. For these experiments, we use only categorical attention heads, with one categorical MLP per-layer. We fix the variable cardinality at 32 and perform a grid search over number of embedding variables, layers, and heads. The embedding parameters are initialized with 300-dimensional GloVe embeddings [Pennington et al., 2014]. We compare the Transformer Program with a standard Transformer, also initialized with GloVe embeddings with grid search of model dimension and number of layers and heads. More details are provided in Appendix §B.3.

| Model | D | L | H | *Precision* | *Recall* | *F1* |
|---|---|---|---|---|---|---|
| Unigram baseline | - | - | - | 82.8 | 55.1 | 66.2 |
| Standard Transformer | 128 | 3 | 4 | 80.4 | 62.7 | 70.5 |
| Transformer Program | 32×4 | 2 | 8 | 81.0 | 71.8 | 76.1 |

Table 2: Named entity recognition on the CoNLL 2003 shared task. For each method, we report the embedding dimension (D), number of layers (L), number of heads (H) of the best-performing model, and precision, recall and F1. For the Transformer Program, the embedding dimension is equal to the variable cardinality multipled by the number of input variables. The unigram baseline predicts the class most commonly associated with each word in the training set.

---

[3]The exceptions are the histogram-related tasks. This could be because the built-in numerical attention mechanism provides an inductive bias for learning histograms.

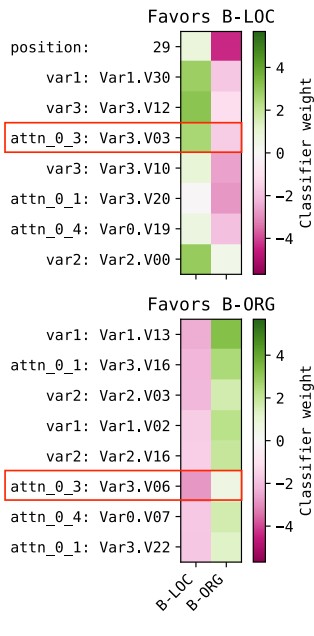

Favors B-LOC

| | | B-LOC | B-ORG |
|---|---|---|---|
| position: | 29 | | |
| var1: | Var1.V30 | | |
| var3: | Var3.V12 | | |
| attn_0_3: | Var3.V03 | | |
| var3: | Var3.V10 | | |
| attn_0_1: | Var3.V20 | | |
| attn_0_4: | Var0.V19 | | |
| var2: | Var2.V00 | | |

Favors B-ORG

| | B-LOC | B-ORG |
|---|---|---|
| var1: Var1.V13 | | |
| attn_0_1: Var3.V16 | | |
| var2: Var2.V03 | | |
| var1: Var1.V02 | | |
| var2: Var2.V16 | | |
| attn_0_3: Var3.V06 | | |
| attn_0_4: Var0.V07 | | |
| attn_0_1: Var3.V22 | | |

(a) Feature weights.

```
# attn_0_3: Copy var3 from previous token
def predicate_0_3(q_position, k_position):
    if q_position in {2}:
        return k_position == 1
    if q_position in {3}:
        return k_position == 2
    if q_position in {4}:
        return k_position == 3
    if q_position in {5}:
        return k_position == 4
    if q_position in {6}:
        return k_position == 5
    # ...
attn_0_3_pattern = select_closest(positions, positions, predicate_0_3)
attn_0_3_outputs = aggregate(attn_0_3_pattern, var3_embeddings)
```

(b) Code for the attention features.

```
class Var3(Enum):
    V00 = ['German', 'television', 'Foreign', 'newspaper', ...]
    V01 = ['<unk>', 'Johnson', 'Morris', 'Service', ...]
    V02 = ['', '', 'Bank', 'York', 'Commission', ...]
    V03 = ['at', 'AT', 'In', 'Saturday', 'match', 'At', ...]
    V04 = ['/', 'up', 'no', 'newsroom', 'Attendance', ...]
    V05 = ['during', 'leader', 'quoted', 'manager', 'came', ...]
    V06 = ['Akram', 'TORONTO', 'BALTIMORE', 'BOSTON', ...]
    V07 = ['said', '"'s', 'has', '@th', 'other', 'shares', ...]
    V08 = ['second', 'told', 'b', 'did', 'spokesman', ...]
    V09 = ['Australia', 'France', 'Spain', 'England', ...]
    V10 = ['Netherlands', 'Finland', 'countries', 'Kurdish', ...]
    # ...
```

(c) The most common words assigned to different values of the `Var3` embedding variable.

Figure 7: We examine how the program distinguishes location entities (`B-LOC`) from organization entities (`B-ORG`) by examining the feature weights with the largest gap between the two classes (7a). Many of the top features are components of the word embeddings, but the model also learns to use the attention heads to gather information from the context. For example, attention head `attn_0_3` copies one of the embedding variables from the previous position (7b). It promotes the `B-LOC` label if the value is `Var3.V03`, which includes prepositions like "at" and "In" (7c). We represent the embedding variables as Python `Enum` objects, which facilitates analysis using a debugger.

**Results.** The results are in Table 2. We compare the best-performing Transformer Program with a standard Transformer and with a unigram baseline, which predicts the tag most frequently assigned to each word in the training data. The Transformer Program achieves reasonable performance, on par with the standard Transformer. In particular, the Transformer Program surpasses the unigram baseline, demonstrating that the method learns to make use of contextual information to predict tags.

**Interpretability.** We illustrate the resulting program by examining how it distinguishes between two commonly confused classes, location entities (`B-LOC`) and organizations (`B-ORG`). To see how the model makes this distinction in general, we extract the linear classifier weights and identify the features with the largest difference between the two classes (Figure 7). The highest ranked features include word-level features (that is, components of the word embeddings); position information; and features computed by the attention heads. Working backwards through the program, we find that the model copies information from the neighboring words—for example, increasing the likelihood of `B-LOC` if the word is preceded by a preposition like "at" or "In".

## 5 Related work

**Learning programs.** Our work has precedent in a variety of existing work on program induction and neuro-symbolic methods [e.g. Reed and De Freitas, 2015, Cai et al., 2017, Andreas et al., 2016, Inala et al., 2020, Cranmer et al., 2020, Kim, 2021]. In particular, a long line of work on Inductive Logic Programming [Muggleton and De Raedt, 1994, Cropper and Dumančić, 2022] has sought to learn symbolic logical programs from data, and a number of recent works have used neural networks to search for discrete logical expressions using differentiable reparameterizations [Payani and Fekri, 2019, Wang et al., 2021, Petersen et al., 2022]. We differ from these methods in targeting programs for the Transformer and focusing on sequence modeling problems.

**Transformers and formal languages.** In addition to RASP [Weiss et al., 2021], a body of research has explored the connection between Transformers and formal languages. Much of this has aimed to

formally characterize the expressivity of Transformers with hard attention [Hahn, 2020, Merrill et al., 2022, Hao et al., 2022]. Giannou et al. [2023] show how Transformers can act as programmable computers by designing a Transformer that can execute programs written in a single-instruction language. Another line of work has attempted to extract deterministic automata, or rules, from neural networks, largely focusing on recurrent neural networks [Jacobsson, 2005, Wang et al., 2018, Weiss et al., 2018]. Merrill and Sabharwal [2022] show theoretically that any fixed-precision Transformer can be expressed as a formula in a type of first-order logic. In contrast to this work, our goal is to design a more interpretable Transformer, rather than extract rules from an existing network.

**Interpretable machine learning models.** Some prior work has introduced architecture changes aimed at making Transformers more interpretable, including sparse attention [Zhang et al., 2021] and changes to the activation function [Elhage et al., 2022]. These methods make some components of the Transformer qualitatively easier to understand; in contrast, our method results in a model that can be fully described by a discrete program. More generally, a growing body of work has sought to develop intrinsically interpretable machine learning models [e.g. Wang and Rudin, 2015, Chen et al., 2019, Rudin, 2019], a motivation we share in this work.

## 6    Conclusion and discussion

We introduced a method for training Transformers that can be deterministically mapped to discrete, human-readable programs. We showed that our method is capable of learning effective solutions for a several synthetic and real-world tasks, and that the resulting programs are easy to interpret. In particular, converting Transformers into programs offers practical benefits—for example, we can use standard code-analysis tools to debug errors. On the other hand, learning discrete programs introduces considerable modeling and optimization challenges. We conclude by mentioning some limitations of our method, which we believe represent promising avenues for future work.

**Scaling challenges.** While we were able to learn a number of effective Transformer Programs, we found that they can struggle to learn on longer inputs, and to learn parsimonious solutions. Are these errors because the framework is not expressive enough, or are they due to optimization issues? Our results suggest that the main issue is optimization. This is evident from the fact that, for many tasks, we can write a Transformer Program that would achieve perfect accuracy, but our method fails to learn these solutions. We conduct a more detailed case study of some optimization failures in Appendix C.3. Therefore, future work is needed to develop better methods for discrete optimization. Additionally, our experiments use a restricted set of modules that may be less expressive than standard Transformers. For example, hard-attention Transformers are less expressive than standard Transformers [Hahn, 2020, Hao et al., 2022]. These restrictions can be relaxed in part by introducing new modules within this framework. We discuss some extensions in Appendix A.4, but leave this direction to future work.

**Are the programs interpretable?** We have argued that our discrete Transformer Programs are more interpretable than standard Transformers, but one might question whether programs are actually easier to understand, especially as we learn larger programs for more complex tasks. On one hand, these programs are interpretable in the sense that we can trace how information flows between different positions and model components, and we can inspect the program using off-the-shelf code analysis tools, like debuggers. For more complex tasks, the program can be understood as a collection of individually interpretable feature functions, rather than a single interpretable algorithm (as illustrated in Figure 7). Nonetheless, the learned programs can still be complicated and non-intuitive. Another avenue for future work is to explore methods for automatically analyzing the resulting programs, and for imposing an inductive bias in favor of more interpretable programs.

**Acknowledgments** We thank the members of the Princeton NLP group for their helpful feedback and discussion. This research is funded by the National Science Foundation (IIS-2211779) and a Sloan Research Fellowship.

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

# A  Method details

## A.1  Categorical attention

As described in Section 3.2, we implement categorical attention by associating each attention head with a boolean predicate matrix, $\mathbf{W}_{\text{predicate}} \in \{0,1\}^{k \times k}$, where $k$ is the variable cardinality, with the constraint that each row $\mathbf{W}_{\text{predicate},i}$ sums to one. The self-attention pattern is then defined by a score matrix $\mathbf{S} \in \{0,1\}^{N \times N}$, with $\mathbf{S} = \mathbf{x}\mathbf{W}_Q\mathbf{W}_{\text{predicate}}(\mathbf{x}\mathbf{W}_K)^\top$. To ensure that each query token attends to a single key token, we use hard attention, defining the attention pattern at position $i$ as $\mathbf{A}_i = \text{One-hot}\left(\arg\max_j \mathbf{S}_{i,j}\right)$. During training, we define $\mathbf{A}_i = \text{Gumbel-Softmax}(\mathbf{S}_i)$.

**Defaulting to the beginning of sequence.** We implement hard attention so that, in the event that there is no matching key for a query, the model defaults to attend to the first token in the sequence. Let $\mathbf{S} \in \{0,1\}^{N \times N}$ denote the score matrix for a sequence with length $N$. We define a modified score matrix $\bar{\mathbf{S}}$ such that, for each row $i$,

$$\bar{\mathbf{S}}_{i,j} = \begin{cases} \mathbf{S}_{i,j} + (1 - \max_j \mathbf{S}_{i,j}) & \text{if } j = 1 \\ \mathbf{S}_{i,j} & \text{otherwise.} \end{cases}$$

**Breaking ties.** Furthermore, we implement the attention mechanism so that, in the event that there is more than one matching key, the model attends to the closest match. Given the score matrix $\mathbf{S} \in \{0,1\}^{N \times N}$, we define the modified score matrix $\bar{\mathbf{S}}$ such that, for each row $i$, $\bar{\mathbf{S}}_{i,j} = \mathbf{S}_{i,j} \times b_{i-j}$, where $b_{i-j} \in [0,1]$ is a bias associated with the offset between position $i$ and $j$. For most experiments, we fix the bias to decrease from 1, when $|i - j| = 1$, to $1/N$, when $|i - j| = N$, with $b_0 = 1/N$ to bias the query at position $i$ against attending to itself. This is similar to types of relative positional bias that have been proposed for regular Transformer-based language models [Press et al., 2022].

## A.2  Additional modules

**Numerical attention.** We implement limited support for numerical variables, designed to ensure that all variables within the program are integers within a bounded range, which allows us to discretize the program by enumerating all possible inputs. First, we include numerical attention heads, which read categorical variables as key and query, and numerical variables as value. The numerical attention head computes attention scores $\mathbf{S} \in \{0,1\}^{M \times N}$ using the `select` operator. Unlike in categorical attention, each query can attend to more than one key, and queries can attend to nothing if there is no matching key. Given attention scores $\mathbf{S} \in \{0,1\}^{M \times N}$ and value variable `var`, the output for the $i^{th}$ token is defined as $\sum_{j=1}^{N} \mathbf{S}_{i,j}\text{var}[j]$. At the input layer, there is a single numerical variable, `ones`, which is frozen and equal to 1 for all positions; attention heads that read `ones` as the value variable are equivalent to the `selector_width` primitive in RASP. At higher layers, numerical attention heads can read the output of lower-layer attention heads as values. Figure 8 depicts a numerical attention head from a program for the Double Histogram task.

Numerical attention does not directly correspond to attention in a standard Transformer, because it computes a weighted sum rather than a weighted average. But a numerical attention head can be implemented in a standard Transformer by composing an attention head with a feed-forward layer, using the beginning-of-sequence token to allow the model to attend to nothing. This is how the `selector_width` primitive is implemented in Tracr [Lindner et al., 2023].

**Feed-forward layers.** In RASP, feed-forward layers are used to implement arbitrary element-wise operations, and they play an important role in many human-written RASP programs. In our Transformer Programs, we restrict the capacity of the feed-forward layers to ensure that they can be decompiled. Each feed-forward layer reads $\ell$ input variables, which are designated in advance to be either numerical or categorical variables, and outputs one new categorical variable. We convert feed-forward layers to programs by enumerating all possible inputs and forming a lookup-table. For all experiments, we set $\ell = 2$. For categorical attention, given variable cardinality $k$, there are $k^2$ possible inputs. An example of a feed-forward layer is given in Figure 9, which depicts a circuit in a program for the Dyck-2 task.

For numerical attention, we can bound the input range by determining the cardinality of the numerical attention heads. For each numerical attention head, the minimum output value is equal to 0 and the maximum output value is equal to the maximum input length multiplied by the cardinality of the input variable (that is, the value variable). For example, if an attention head reads `ones` as the value

```python
def num_predicate_0_1(q_token, k_token):
    if q_token in {"0"}:
        return k_token == "0"
    elif q_token in {"1"}:
        return k_token == "1"
    elif q_token in {"2"}:
        return k_token == "2"
    elif q_token in {"3"}:
        return k_token == "3"
    elif q_token in {"4"}:
        return k_token == "4"
    elif q_token in {"5"}:
        return k_token == "5"
    elif q_token in {""}:
        return k_token == "<pad>"

num_attn_0_1_pattern = select(
    tokens, tokens, num_predicate_0_1)
num_attn_0_1_outputs = aggregate_sum(
    num_attn_0_1_pattern, ones)
```

```python
def num_mlp_0_1(num_attn_0_1_output):
    key = num_attn_0_1_output
    if key in {0, 1}:
        return 4
    return 0

num_mlp_0_1_outputs = [
    num_mlp_0_1(k0)
    for k0 in num_attn_0_1_outputs]
```

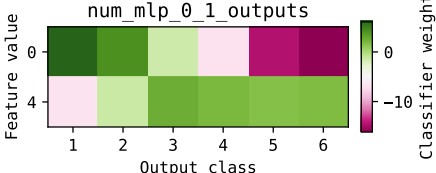

Figure 8: An example of a numerical attention head and MLP in a program for the Double Histogram task. For this task, the model must output, for each position, the number of unique tokens with the same histogram value. In this example, an attention head (*left*) calculates the histogram for each position. An MLP (*top right*) reads the histogram values and outputs a value of 0 if the histogram value is greater than one, and 4 otherwise. Inspecting the corresponding classifier weights (*bottom right*), we see that an output value of 0—meaning a histogram count greater than 1—increases the likelihood that the double-histogram value is 1 or 2, and decreases the likelihood of larger values. Because the input length is limited to 8, this reflects the fact that if one number appears many times, it is unlikely that another number appears the same number of times. An output of 4 (meaning a histogram count of 1) increases the likelihood that the double-histogram is greater than 1. Note that we configure all MLPs to read two input variables, but some MLPs learn to read the same variable for both inputs, as in this example. This allows us to compress the corresponding function.

variable, the maximum output value is equal to the maximum input length. If an attention head reads the output of a first-layer attention head as value, the maximum output value is equal to the square of the maximum input length.

### A.3 Extracting programs

In this section, we provide more details about our procedure for converting our trained models into Python programs. While there are many possible ways to express the discretized model as Python code, we choose a mapping that facilitates analysis with a standard Python debugger. We use several simple strategies to improve the readability of the code, by annotating variable types, compressing statements, and removing unreachable branches. Illustrative programs are depicted in Figures 8 and 9.

**Attention.** Each attention head is represented by a `predicate` function, which takes as input a key and query and outputs a value in $\{0, 1\}$. In Transformer Programs, all keys and queries are categorical variables with cardinality $k$, so the `predicate` function can be defined by enumerating the possible query values, which we do with a series of `if` statements. Additionally, if multiple query values are mapped to the same key, we condense the `predicate` by combining these queries into a single branch. This is illustrated in Figure 9. In the first attention head (*left*), each query position attends to the previous position (with the exception of the first token), so we cannot apply any compression. In the second attention head (*bottom right*), fifteen out of the sixteen possible query values are mapped to a single key value, so we combine them into a single branch.

**MLPs.** We convert feed-forward modules to functions by enumerating all possible inputs and forming a key/value lookup-table. For all experiments, we set each MLP to read $\ell = 2$ input variables. In some cases, the MLP learns to read the same variable for both input variables, allowing us to reduce the length of the function. We further reduce the length of these functions by identifying the MLP output value associated with greatest number of keys, and returning this value as the default value without explicitly evaluating the corresponding condition. These two forms of compression are illustrated in Figure 8 (*top right*). This MLP reads a single numerical input variable and outputs a value of 4 if the input is 0 or 1 and a value of 0 otherwise.

```python
# First attention head: copy previous token.
def predicate_0_0(q_position, k_position):
    if q_position in {0, 13}:
        return k_position == 12
    elif q_position in {1}:
        return k_position == 0
    elif q_position in {2}:
        return k_position == 1
    elif q_position in {3}:
        return k_position == 2
    elif q_position in {4}:
        return k_position == 3
    elif q_position in {5}:
        return k_position == 4
    elif q_position in {6}:
        return k_position == 5
    elif q_position in {7}:
        return k_position == 6
    elif q_position in {8}:
        return k_position == 7
    elif q_position in {9}:
        return k_position == 8
    elif q_position in {10}:
        return k_position == 9
    elif q_position in {11}:
        return k_position == 10
    elif q_position in {12}:
        return k_position == 11
    elif q_position in {14}:
        return k_position == 13
    elif q_position in {15}:
        return k_position == 14
attn_0_0_pattern = select_closest(positions, positions,
                                  predicate_0_0)
attn_0_0_outputs = aggregate(attn_0_0_pattern, tokens)
```

```python
# MLP: reads current token and previous token
# Outputs 13 if it sees "(}" or "{)".
def mlp_0_0(token, attn_0_0_output):
    key = (token, attn_0_0_output)
    if key in {(")", ")"),
               (")", "}"),
               ("{", ")"),
               ("}", ")"),
               ("}", "}")}:
        return 4
    elif key in {(")", "{"),
                 ("}", "(")}:
        return 13
    elif key in {("(", ")"),
                 ("(", "}"),
                 (")", "("),
                 ("{", "}"),
                 ("}", "{")}:
        return 0
    return 7
mlp_0_0_outputs = [
    mlp_0_0(k0, k1) for k0, k1 in
    zip(tokens, attn_0_0_outputs)
]

# 2nd layer attention: check for "(}" or "{)"
def predicate_1_2(position, mlp_0_0_output):
    if position in {0, 1, 2, 4, 5, 6, 7, 8, 9,
                    10, 11, 12, 13, 14, 15}:
        return mlp_0_0_output == 13
    elif position in {3}:
        return mlp_0_0_output == 4
attn_1_2_pattern = select_closest(
    mlp_0_0_outputs, positions, predicate_1_2)
attn_1_2_outputs = aggregate(
    attn_1_2_pattern, mlp_0_0_outputs)
```

Figure 9: An example of a circuit in a program for Dyck-2. For this task, the inputs consist of strings from the vocabulary $\{(,),\{,\}\}$. At each position $i$, the model must output T if the string up to position $i$ is a valid string in Dyck-2; P if it is the prefix of a valid string; and F otherwise. A string is valid if every parenthesis is balanced by a parenthesis of the same type. This circuit recognizes an invalid pattern, where a left parenthesis (( or {) is immediately followed by a right parenthesis of the wrong type (} or ), respectively). First, an attention head (*left*) copies the tokens variable from the previous position. Second, an MLP (*top right*) reads the output of this attention head, along with the tokens variable, and classifies the input into one of four categories—in particular, returning 13 if it sees the pattern (} or {). In the second layer, another attention head (*bottom right*) looks for 13, propagating this information to later positions.

**Annotating variable types.** In our Transformer model, all categorical variables are encoded using one-hot embeddings, which are represented as integers in the corresponding program. To improve readability, we replace these integers with symbolic values where appropriate. At the input layer, we represent the values of the tokens variable as strings rather than integer indices. At subsequent layers, we determine the appropriate type by following the computation graph. For example, in Figure 9, the tokens variable takes on values in $\{(,),\{,\}\}$; the first attention head reads tokens as the value variable, so we can automatically determine that attention_0_0_outputs variable takes on values of the same type; finally, the MLP reads tokens and attention_0_0_outputs as input variables, so we define the mlp_0_0 function in terms of the token value type.

## A.4 Possible extensions

In our experiments, we used a limited set of model components that were designed to ensure that the resulting programs are interpretable. However, these do not cover all of the primitive components included in RASP. In this section, we discuss some of these differences; how these limitations can be addressed by introducing additional modules; and the relationship between (a) introducing more expressive modules and (b) the complexity of the corresponding program.

**One-to-many attention predicates** Our Transformer Programs use a form of binary categorical attention that associates each query value with a single key value. This is enforced by parameterizing the attention head with a predicate matrix $\mathbf{W}_{\text{predicate}} \in \{0,1\}^{k \times k}$ (where $k$ is the variable cardinality) and constraining $\mathbf{W}_{\text{predicate}}$ so that each row contains only a single non-zero entry. This choice results

in more concise programs, as each attention head can be characterized in terms of a one-to-one mapping between key and query values, but it also makes it more difficult to express certain common attention patterns. For example, it is difficult to implement the "less than" predicate, by which a query with a value of $n$ attends to all keys with values $< n$. This predicate is used in a number of RASP solutions [Weiss et al., 2021] to implement a sorting routine. It is straightforward to allow the predicate matrix to have multiple non-zero entries per row. These weights could be learned by replacing the row-wise Gumbel softmax operation described in §3.2 with an entry-wise Gumbel-sigmoid operation. In preliminary experiments, we trained models with this form of predicate matrix and found they obtained similar performance but were qualitatively more difficult to understand.

**Score-based attention**  As in Tracr and RASP, we require that attention heads with categorical values always attend to exactly one position (hard attention). We enforce this constraint by deterministically breaking ties in favor of the position nearest to the query position. Weiss et al. [2021] discuss an extension to RASP (`select_best`) that provides for a more general form of one-hot attention, where each attention head is associated with both a binary function, $\mathrm{predicate} : \mathcal{Q} \times \mathcal{K} \rightarrow \{0, 1\}$, and a continuous function, $\mathrm{score} : \mathcal{Q} \times \mathcal{K} \rightarrow \mathbb{R}$, that maps each combination of key and query to a real value. Given a query $q$ and keys $k_1, \ldots, k_n$, the attention between $q$ and $k_i$ is defined as 1 if both $\mathrm{predicate}(q, k_i) = 1$ and $\mathrm{score}(q, k_i) = \max_{j=1}^{n} \mathrm{score}(q, k_i)$, and 0 otherwise. This type of operation has a number of applications. For example, Yao et al. [2021] describe a construction for modeling sequences of balanced parentheses with an autoregressive Transformer, in which the model attends to the *most recent* opening bracket at the same nesting depth as the current position, by defining $\mathrm{score}(q, k_i) = i$.

It would be relatively straightforward to extend Transformer Programs to include score-based attention modules, but this would also result in longer and more complex programs, because each query must now be defined in terms of an ordering of all $|\mathcal{K}|$ possible keys, instead of a single key. To see why this makes the program more difficult to understand, consider the information gained by observing that some token $i$ attends to token $j$. In our default, one-to-one form of attention, this observation means that $j$ is the closest position such that $\mathrm{predicate}(q_i, k_j) = 1$. In score-based attention, the observation also implies that there is no position $\ell$ such that $\mathrm{score}(q_i, k_\ell) > \mathrm{score}(q_i, k_j)$, conveying additional information about the rest of the sequence.

**Learnable numerical variables**  In our experiments, we constrain the numerical variables to be either constant and equal to one (at the input layer), or the output of a numerical attention head. In RASP, numerical variables can take on any scalar value. Transformer Programs could be extended to include learnable numerical input variables, at the cost of some interpretability tradeoffs. For example, it may no longer be possible to decompile MLPs by enumerating all possible arguments.

# B  Experiment details

In this section we describe additional implementation details for the experiments in Section 4. Code to reproduce the experiments is available at https://github.com/princeton-nlp/TransformerPrograms.

## B.1  In-context learning

**Data.**  For our in-context learning experiment (Section 4.1), we sample 20,000 sequences of length 10. The first token is a beginning-of-sequence token, and the remainder of the sequence is formed by randomly sampling a many-to-one mapping between letter types (either a, b, c, d) and numbers (0, 1, 2, 3), and then alternating letters and numbers until reaching the target length. The model is trained to predict the number following each letter, or a special unk token if the letter has not appeared earlier in the sequence. We use an autoregressive mask so that at each position $i$, the model can attend only to keys with positions $j \leq i$.

**Training.**  We train each model for 250 epochs with a batch size of 512, a learning rate of 0.05, and annealing the Gumbel temperature geometrically from 3.0 to 0.01, decreasing the temperature at each training step. We take one Gumbel sample per step. The model has two layers with one categorical attention head per layer, with a variable cardinality of 10. We train five models with different random initializations and pick the one with the lowest test loss. We implement all models in PyTorch [Paszke et al., 2019] and use the Adam optimizer [Kingma and Ba, 2014].

## B.2 RASP tasks

**Data.** For each RASP task, we sample 20,000 inputs without replacement and partition them into train, validation, and test sets containing 16,000/2,000/2,000 instances respectively. With the exception of the Dyck tasks, we sample inputs by sampling tokens uniformly from the vocabulary. For the Dyck tasks, we follow Weiss et al. [2021] and sample with a bias towards strings with a valid prefix. Specifically, given maximum length of $N$, with probability 0.5, we sample $N$ tokens uniformly from the vocabulary; otherwise, we sample a valid Dyck string $s$ with length $|s| \leq N$, and append $N - |s|$ randomly sampled tokens to the end to obtain a string with length $N$. For all tasks, we obtain 20,000 samples without replacement by sampling strings uniformly until the set of unique strings has the intended size. We prepend all inputs with a beginning of sequence token bos. For the sort and reverse tasks, we also append an end-of-sequence token, eos. Following Weiss et al. [2021], we report the token-level accuracy.

**Training.** As above, we train the model for 250 epochs with a batch size of 512, and a learning rate of 0.05. We anneal the Gumbel temperature geometrically from 3.0 to 0.01, decreasing the temperature at each training step, and taking one Gumbel sample per step. These hyperparameters were chosen after initial experiments on the RASP tasks. We do a grid search for number of layers (2, 3), number of attention heads (4, 8), and number of MLPs per layer (2, 4). The attention heads are evenly divided between categorical and numerical heads. Simililarly, the MLPs are evenly divided between MLPs with two categorical inputs, and MLPs with two numerical inputs. We train models with five random initializations, pick the model with the best accuracy on a validation set, and report the accuracy on a held-out test set. For experiments with standard Transformers (described in Appendix C.1), we similarly do a grid search for number of layers (2, 3) and attention heads (4, 8) but otherwise follow the hyper-parameters described by Weiss et al. [2021]: the hidden dimension is 256, the learning rate is 0.0003, the batch size is 50, and we train for up to 100 epochs, picking the model checkpoint from the epoch with the highest validation accuracy. Each model takes between five and fifteen minutes to train on an Nvidia RTX 2080 GPU, depending on the number of layers.

## B.3 Named entity recognition

**Data.** For our named entity recognition experiments (Section 4.3), we train on data from the CoNLL-2003 shared task [Sang and De Meulder, 2003], using the distribution from HuggingFace Datasets [Lhoest et al., 2021]. This data is pre-tokenized and we filter the dataset to sentences with a maximum length of 30 tokens and add special beginning- and end-of-sequence tokens. Following Collobert et al. [2011], we preprocess the data by replacing all contiguous sequences of numbers with a special number symbol, so, for example, the string "19.99" is replaced with "#.#". We use the standard train/validation/test split and evaluate the results using a Python implementation of the standard CoNLL evaluation script [Nakayama, 2018].

**Training.** For both the standard Transformer and Transformer Programs, we use a batch size of 32 and perform a grid search over the number of layers (1, 2, or 3) and the number of attention heads (4, 8). For the standard Transformer, we search for model hidden dimension (64, 128, 256) and train for up to 100 epochs, taking a checkpoint after each epoch and picking the checkpoint with the highest performance on the validation set. For the Transformer Program, we fix the variable cardinality to 32 and search for the number of embedding variables (2, 4, 8) and training epochs (30, 50, 100). As above, we anneal the temperature geometrically from 3.0 to 0.01, and we report results after discretizing the model at the end of training. For both models, we initialize the word embeddings using 300-dimensional GloVe embeddings For both models, we train with three random seeds, pick the model with the highest F1 score on the validation set, and report the results on the test set.

# C Additional results

## C.1 Additional RASP results

In this section, we provide additional code examples and ablations on the RASP tasks.

**Larger-scale versions of the tasks.** In Figure 10, we show the accuracy of Transformer Programs trained on RASP tasks with a longer maximum sequence length and a larger input vocabulary. Because we use one-hot encodings for the token and position embeddings, these values determine the cardinality, $k$, of the categorical variables used in the program. We compare Transformer Programs with standard Transformers, for each model performing a grid search over the number of layers and

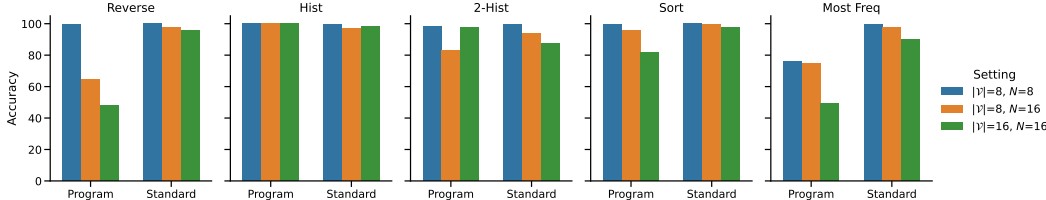

Figure 10: RASP accuracy after increasing the size of the input vocabularies ($|\mathcal{V}|$) and maximum sequence length ($N$), comparing Transformer Programs with standard Transformers. For each model, we perform a hyperparameter search over the number of layers and attention heads and report the test accuracy of the model that performs best on the validation set. On most tasks, performance degrades moderately when trained on longer sequences, and degrades more when we also increase the vocabulary size, and this trend generally holds for both standard Transformers and Transformer Programs. However, the Transformer Programs deteriorate more dramatically, with over a 50% accuracy gap relative to the standard Transformer in the most difficult setting.

attention heads and report the test accuracy of the model that performs best on the validation set. In Figure 10, we increase the vocabulary size and maximum sequence length from eight to sixteen. All other experiment details are the same as in Appendix B.2. Performance degrades on longer sequences, underscoring some of the optimization challenges in learning Transformer Programs for larger scales.

**No numerical variables.** In our main experiments, we train Transformer Programs with an equal number of categorical attention heads and numerical attention heads, and categorical MLPs and numerical MLPs. In Table 3, we compare results on RASP tasks with only categorical variables. The experiment setting is otherwise the same as in Appendix B.2, but all attention heads and MLPs are constrained to read only categorical variables. Not surprisingly, performance degrades on three tasks that are primarily based on counting: Histograms, Double Histograms, and Most Frequent. However, the Transformer Programs still achieve good performance on the other four tasks. These include the balanced parenthesis languages (Dyck-1 and Dyck-2), which are most naturally solved by keeping a count of unmatched parentheses at each position.

| Dataset | L | H | M | Acc. |
|---|---|---|---|---|
| Reverse | 3 | 8 | 4 | 99.75 |
| Hist | 3 | 8 | 4 | 83.20 |
| 2-Hist | 2 | 8 | 2 | 55.89 |
| Sort | 3 | 2 | 2 | 99.99 |
| Most Freq | 3 | 8 | 4 | 64.13 |
| Dyck-1 | 3 | 8 | 4 | 99.43 |
| Dyck-2 | 3 | 4 | 2 | 99.17 |

Table 3: Accuracy of Transformer Programs on RASP tasks using only categorical variables, with the number of layers (L), attention heads (H), and MLPs (M) used in the best-performing model.

### C.2 Analyzing the generated code

Here, we provide some additional analysis of the generated code, focusing on the RASP tasks. Complete, generated programs are available at https://github.com/princeton-nlp/TransformerPrograms.

**Code analysis tools.** One advantage of representing models as Python programs is that we can use off-the-shelf code analysis tools to interpret them. Figure 11 illustrates one example, showing how we can use a Python debugger to understand a non-trivial circuit in a Sort program.

**Program length.** Table 4 shows the number of lines in our best-performing program for each RASP task, before and after applying the compression strategies described in Appendix A.3. The program length includes the basic library functions used to We use an automated Python code formatter,[4] which applies a number of standard style conventions and, in particular, enforces a maximum line length of 88 characters.

---

[4] https://github.com/psf/black

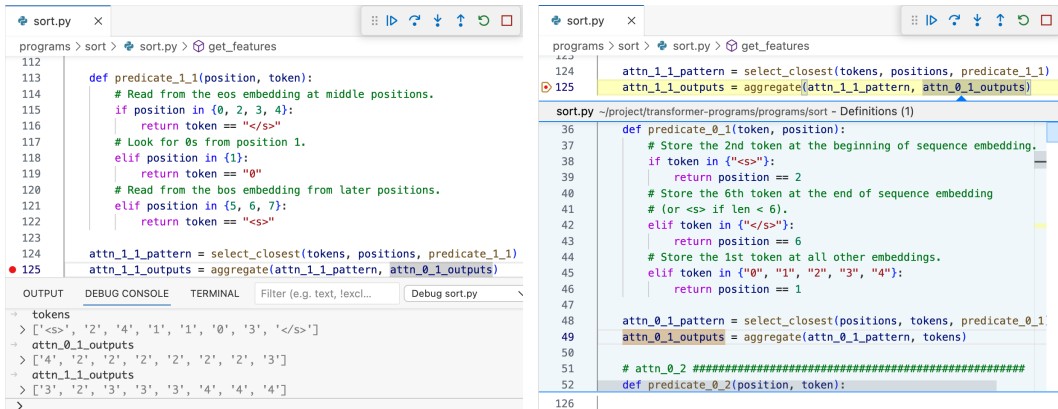

Figure 11: After converting a Transformer Program into Python code, we can analyze it using off-the-shelf debugging tools. Here, we inspect a subset of the Sort program using the built-in debugger in Visual Studio Code. We can step through the program for a test example, set break-points, and leave comments. In this example, we find a circuit involving two attention heads, with a second-layer attention head (left) reading a value from a first-layer attention head (right). Inspecting the code, we can see that this pair of heads has the effect of propagating an early-position token to the later positions and a late-position token to the earlier positions.

| Dataset | Full | Pruned |
|---------|------|--------|
| Reverse | 1893 | 713 |
| Hist | 324 | 160 |
| 2-Hist | 1309 | 423 |
| Sort | 1503 | 635 |
| Most Freq | 1880 | 666 |
| Dyck-1 | 9975 | 892 |
| Dyck-2 | 5406 | 733 |

Table 4: The number of lines in best programs for each RASP task before and after applying a set of simple pruning strategies based on static analysis of the code.

**What information do the attention heads read?** Because each attention head reads a fixed set of named variables, we can characterize how information flows through the programs by examining which variables are read by each head. In Figure 12, we summarize this information for RASP programs. At the first layer, the majority of categorical attention heads read `positions` as key and query variables and `tokens` as the value. At higher layers, `positions` remains the most common key variable, but the models are more likely to read the outputs of lower-layer attention heads as the value variable. Numerical attention heads are less likely to read `positions` and more likely to read `tokens`, `attn`, and `mlp` outputs. Both kinds of attention successfully learn to compose modules, with higher-layer modules reading the outputs of modules at lower layers.

### C.3 Optimization challenges: case study

What are the challenges to scaling Transformer Programs to more complex problems? As discussed in Section 6, some of the main obstacles are optimization challenges. This is evident from the fact that, for many RASP tasks, we can write a Transformer Program that would achieve perfect accuracy, but our method fails to learn these solutions (as illustrated in Fig. 10). In this section, we conduct a case study to better understand these optimization challenges.

**Setting** We focus on the *Reverse* task. This task can be solved by the following program, adapted from Weiss et al. [2021], using a Transformer with two layers and one attention head per-layer:

```
# First-layer attention
length = aggregate(select(tokens, tokens, lambda q, k: k == ""), positions)

# First-layer MLP
targets = one_hot(length - positions)
```

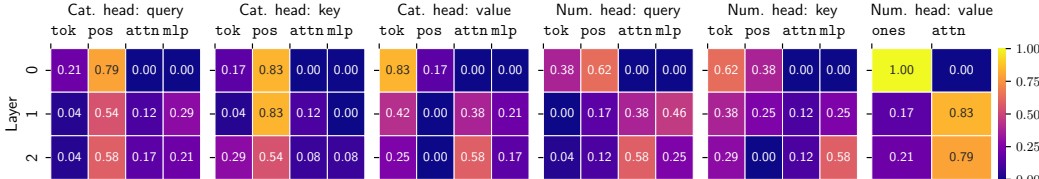

Figure 12: For each of the RASP tasks, we learn a Transformer Program with three layers and eight heads per-layer, divided evenly between categorical and numerical attention heads, and summarize the types of variables that are read at different layers. For each layer, we list the key, query, and value variables read by attention heads at that layer, and calculate the proportion of heads that read the `tokens` variable; `positions`; `ones` (for numerical attention); the output of a previous attention head (`attn`); or the output of a previous MLP (`mlp`). We aggregate over RASP programs and compare categorical attention heads (*left*) and numerical attention heads (*right*).

| | **Attention 1** | | **MLP 1** | **Attention 2** | | |
| | *Read* | *Predicate* | *Read* | *Read* | *Predicate* | Accuracy |
|---|---|---|---|---|---|---|
| | - | - | - | - | - | 23.2/23.6/24.1 |
| | ✔ | ✔ | ✔ | ✔ | ✔ | 99.9/99.9/80.8 |
| | ✔ | ✔ | ✔ | ✔ | - | 37.9/40.3/18.5 |
| | ✔ | ✔ | ✔ | - | ✔ | 17.1/13.7/20.2 |
| | ✔ | ✔ | - | ✔ | ✔ | 95.1/94.1/95.3 |
| | ✔ | - | ✔ | ✔ | ✔ | 99.1/83.9/78.2 |
| | - | ✔ | ✔ | ✔ | ✔ | 35.5/44.1/41.8 |

Table 5: Results on the *Reverse* task ($|\mathcal{V}| = N = 16$) after manually initializing model components to encode a generalizing solution. (See Section C.3.) In each row, we choose a subset of components to manually initialize (✔), initialize the other weights randomly (-), and then train the model, reporting the (per-token) accuracy of the resulting program. The components we consider are the projection matrices, which determine which variables each module reads (*Read*), and the attention predicate matrices (*Predicate*). All other components (i.e., the internal MLP parameters and the final classifier weights) are always initialized randomly. We run each experiment with three random seeds and report the accuracy from each run. When we manually initialize the attention *Read* and *Predicate* weights to encode the generalizing solution, the model successfully learns the remaining components. Performance degrades considerably when even a single attention components is initialized randomly, suggesting that our optimization procedure struggles to find effective attention patterns.

```
# Second-layer attention
output = aggregate(select(targets, positions, ==), tokens)
```

The first attention layer copies the position of the end-of-sequence token to determine the length of the sequence; the first-layer MLP calculates the difference between `length` and `positions`; and the second attention layer uses the MLP output to read, at each position $i$, the token at position $\texttt{length} - i$. This solution works for sequences of all lengths, but Transformer Programs evidently fail to learn it.

**Method** To better understand why, we manually initialize some components of a two-layer Transformer Program model to encode this solution and train the model on the *Reverse* task with a vocabulary size and maximum sequence length of 16. Specifically, in the attention layers, we manually initialize: (1) the parameters associated with $\boldsymbol{\pi}_K$, $\boldsymbol{\pi}_Q$, and $\boldsymbol{\pi}_V$, which identify the key, query, and value variables to read; and (2) the parameters associated with the predicate matrix $\mathbf{W}_{\text{predicate}}$. In the MLP, we manually initialize the parameters associated with the input projection matrix $\mathbf{W}_{\text{in}}$, which determines which variables the MLP reads from the residual stream. We do not initialize the other MLP weights or the classifier weights. We initialize different subsets of these parameters and observe whether the model learns the remaining components needed to complete the solution. We train each model for 100 epochs but otherwise use the same hyperparameters described in Appendix B.

**Results** The results of this experiment are in Table 5. When all of the components are randomly initialized, the model fails to learn the generalizing solution. When we manually initialize the attention layers and the MLP input weights, the model learns the remaining components needed to solve the problem successfully. In particular, it finds the internal MLP weights needed to calculate the difference between the two input variables (*length* and *position*) and output a one-hot encoding of the result. Interestingly, if we randomly initialize either even a single component in either one of the attention layers, the model fails to learn the correct solution; but if we randomly initialize only the MLP, the model learns successfully. These results suggest that the attention layers may be more difficult to optimize—for example, they may be more likely to get stuck in local minima.

## C.4 Text classification

| Model | TREC | MR | Subj | AG |
|---|---|---|---|---|
| Bag of words | 74.8 | 77.8 | 92.6 | 89.6 |
| Standard Transformer | 88.6 | 77.2 | 92.3 | 91.7 |
| Transformer Program | 85.6 | 77.9 | 93.0 | 90.8 |

Table 6: Classification accuracy on question classification (**TREC**); sentiment analysis (**MR**); subjectivity classification (**Subj**); and topic classification (**AG news**). The Bag of words baseline is a multinomial Naive Bayes model trained on unigram features.

Next, we train Transformer Programs for three standard text classification datasets: classifying questions as one of six topics [TREC; Voorhees and Tice, 2000]; classifying movie reviews as positive or negative [MR; Pang and Lee, 2005]; classifying sentences as objective or subjective [Subj; Pang and Lee, 2004]; and classifying sentences from news articles as one of four topics [AG News; Zhang et al., 2015]. We filter the datasets to sentences with at most 64 words and use a vocabulary of the 10,000 most common words, replacing the remaining words with an *unknown* token, and fixing the variable cardinality at 64. As above, we compare the Transformer Program with a standard Transformer. For both models, we use the Transformer to extract token embeddings, obtain a sentence embedding by averaging the token embeddings, and train a linear classifier on the sentence embedding. We initialize both models with 300-dimensional GloVe embeddings and use grid search to select the model dimension, number of layers, and number of heads, and training hyper-parameters (learning rate and number of training epochs), as described in Appendix B.3. We hold out 10% of each training set to use as a validation set. For both models, we train with three random seeds, pick the model with the highest accuracy score on the validation set, and report the results on the standard test set.

**Accuracy.** Table 6 compares the accuracy for Transformer Programs, standard Transformers, and a bag-of-words baseline. The Transformer Programs are competitive with the standard Transformer on three out of four datasets, performing slightly worse on TREC. This could be because the standard Transformer can more effectively leverage pre-trained word embeddings, or because it is easier to regularize.

**Interpretability.** We illustrate the interpretability of these programs in Figure 13 by inspecting the features for an example from the TREC dataset.

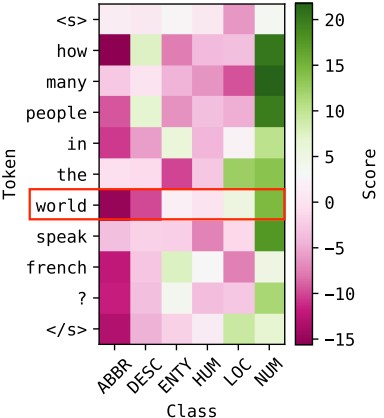

(a) Classification scores for each token embedding.

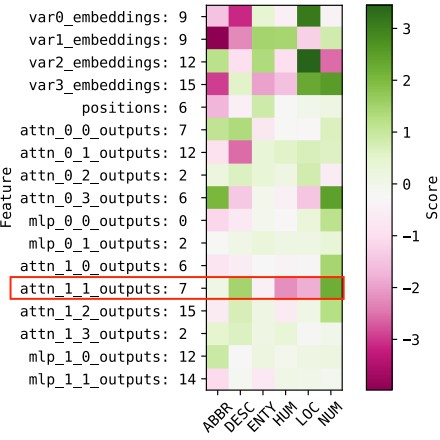

(b) Feature-level scores for the "world" token.

```
# attn_1_1: Copy var0 from early positions.
def predicate_1_1(var2_embedding, position):
    if var2_embedding in {0, 4, 5, 6, 10, 12, 13}:
        return position == 1
    elif var2_embedding in {1, 7, 8, 9, 11, 14}:
        return position == 4
    elif var2_embedding in {2, 3}:
        return position == 5
    elif var2_embedding in {15}:
        return position == 2

attn_1_1_pattern = select_closest(
    positions, var2_embeddings, predicate_1_1)
attn_1_1_outputs = aggregate(
    attn_1_1_pattern, var0_embeddings)
```

(c) The code for computing the attn_1_1_outputs feature.

| word | var0... | var1... | var2... | var3... |
|---|---|---|---|---|
| how | 7 | 4 | 5 | 15 |
| are | 7 | 11 | 4 | 0 |
| for | 7 | 2 | 11 | 10 |
| when | 7 | 6 | 14 | 6 |
| there | 7 | 5 | 14 | 15 |
| year | 7 | 6 | 6 | 14 |
| people | 7 | 13 | 15 | 15 |
| time | 7 | 6 | 15 | 0 |
| date | 7 | 6 | 14 | 6 |

(d) A subset of the embedding table, filtering to words with var0_embedding values of 7.

Figure 13: Visualizing the predictions for an example from the TREC question classification dataset. We classify sequences by pooling the final-layer token embeddings, so we can visualize the classification scores for each token embedding (Figure 13a). In this example, the first three tokens ("how", "many", and "people") have the highest scores in favor of the NUM class, but most other tokens have high scores for this class as well. To see why, we can inspect the feature-level scores for individual token embeddings. In Figure 13b, we display the categorical features of the "world" token along with the corresponding classifier weights. For this token, the input features favor the LOC label, but attention features increase the score for NUM. Figure 13c displays the subset of the program that calculates one of these attention features (attn_1_1_output = 7). This attention head generally copies the var0_embeddings variable from first or fourth position—positions that can be expected to be informative for question classification. In Figure 13d, we display the most frequent words that have var0_embedding values of 7: they include question words like "how" and "when" and nouns like "year", "time", and "date", which are more likely to occur in numerical questions.

