# OpenReview forum: "Learning Transformer Programs"
_NeurIPS.cc/2023/Conference — NeurIPS 2023 oral_

### Official Review · Reviewer_TaJd · 2023-07-05

**Soundness:** 2 fair
**Presentation:** 4 excellent
**Contribution:** 3 good
**Rating:** 6
**Confidence:** 2

**Summary:**

_Background_: RASP is a tensor processing language which provides a language to hand-write transformers. Tracr presents a framework to compile RASP programs to a transformer architecture automatically.

The authors of this project are interested in distilling a transformer architecture to an equivalent program in RASP. This is challenging because RASP is not a sound language: the behavior of many trained transformers is equivalent to that of a single RASP program. The authors go around this problem by constraining the transformer architecture in strategic places so that there is a deterministic mapping from a transformer to an equivalent program in RASP. The authors present a model distillation procedure to construct such reduced transformers (called Transformer Programs) automatically. The authors test the Transformer Programs on three tasks: 1) in-context learning, 2) recovering handwritten RASP programs, and 3) simple real-world NLP tasks. They observe mixed results: transformer programs can fully solve (1), they cannot generalize to larger sequences for (2), and they are better than non-sequential baselines on (3) but worse than generic transformers. The authors also show the utility and interpretability of the distilled programs by analyzing the programs using off-the-shelf debuggers.

**Strengths:**

Originality
* RASP programs are inherently unsound: one RASP program is equivalent to many trained transformers. Solving this problem by constraining the transformer to elicit a bijective mapping is a simple yet elegant idea.

Significance
* This paper shows a simple method to distill transformers into interpretable programs. This work opens the door for work in program synthesis, static analysis, and model checking to be used for sequential processing tasks.
* Many downstream tasks rely on transformers to processing temporal sequences of data into labels. While the transformer architecture presented is constrained, an interpretable transformer can be very useful for tasks where high degree of interpretability and accuracy are required.

Clarity and Quality
* The framework is well motivated, and the writing is clear and precise.

**Weaknesses:**

__Weaknesses__

I have three main concerns (paraphrased from the questions section below):
* I’m concerned that the constraints mentioned here violate properties[1] that the mechanistic interpretability community have identified as being intrinsic to the utility of transformers.
* Poor robustness: Model distillation has been observed to produce models that are more robust to outliers while being slightly less accurate[2]. However, in experiments it seems that the distilled models are neither robust to outliers nor as accurate as baselines, which makes it harder to use this as a drop-in replacement.
* Lack of Reusability: An interesting property of the RASP programs (and RASP distilled transformers) is that many tasks can be efficiently solved by precomputing useful primitives. The algorithm presented here does not seem to model this property. For instance, in the reverse proposed in RASP, the authors' solution precomputes length of the sequence and then uses length to flip the indices of an identity matrix. The derived transformer also captures this behavior. Does the Transformer Program also capture this behavior? If not, why not?

Clarity:
* 135. subpace -> subspace
* Related work: I defer to the authors, however, I feel adding a section on other model distillation approaches would be relevant. In the context of model distillation, [3] seems similar to this work in learning a transformer (for multi-agent communication) and then by discretizing it (using MCMC) to obtain an interpretable policy.

Overall, I've given the paper a __borderline rejection__. The problem statement presented has promise to have significant impact, however, the experimental results do not instill confidence in the downstream utility of this method and I believe this paper will benefit from another round of review. I'm more than willing to change my recommendation after the rebuttal period.

====
__EDIT__: I misinterpreted the paper as a _model distillation_ paper instead of a continuous relaxation one. In light of this, and other clarifications presented by the authors, I'm raising my score to a __weak accept__.

1. https://transformer-circuits.pub/2022/toy_model/index.html
2. https://arxiv.org/abs/2006.11287
3. https://arxiv.org/abs/2101.03238



**Questions:**

* Constraint 1 “the token embeddings encode … fixed set of variables… in orthogonal subspace. What is a variable here? I’m going to assume variable=token, but it would be good to define this.
   * Does this mean that encoding 100 variables requires a 100 dimensional vector? Does this assumption violate some of the properties we know about transformers? For instance: nonlinear superposition[1]?
* 155: “require that each query token attend to a single key token”. This constraint seems very limiting. Do most sequential problems where transformers are used adhere to this constraint?
* 256 “results degrade on longer sequence tasks.” This seems concerning. Are these results for the trained model or for the extracted program? Ideally, model distillations are more robust to OOD tasks than the model itself[2]. Is there any indication for why this might be the case?
. 259 “our solution might be different from human written solutions.” This seems concerning because RASP programs attempt to upper-bound the number of transformer layers that might be needed for computation. Do we have any indication for why this might be happening? Is this maybe because the human written solutions reuse previously computed values?

1. https://transformer-circuits.pub/2022/toy_model/index.html
2. https://arxiv.org/abs/2006.11287


**Limitations:**

The authors have addressed the limitations of the method adequately to the best of my knowledge.

---

> ### Author Rebuttal · Authors · 2023-08-09
>
> Thank you for your review! We believe there is some misunderstanding in the main concerns, and we have addressed them below. We kindly request the reviewer to reconsider the evaluation, and we  are more than happy to further clarify if anything is still unclear.
>
> > Do the constraints violate properties (e.g. superposition)?
>
> Our constraints do prevent the model from using feature superposition. As noted by [1], superposition is a major obstacle to interpretability. Our work is one attempt to solve it. It is not clear that superposition is “intrinsic to the utility of transformers” – in [1], the authors suggest that superposition is useful but might not be necessary; in fact, every model can be seen as equivalent  to a “hypothetical disentangled model” that uses higher-dimension embeddings. On the other hand, we agree that superposition is useful, because it enables the use of lower-dimensional representations. One avenue for future work is to extend our framework to enable the use of more compressed representations, while still preserving interpretability.
>
> Please let us know if there are other properties that raise concerns (aside from superposition).
>
> > Poor robustness / relationship to model distillation
>
> We believe this concern might be based on a misunderstanding:  our work is not really a model distillation approach, or analogous to the method in [2] (which targets a different domain and architecture–physics modeling with GNNs). Our approach is better thought of as a continuous relaxation of discrete search, where the search space is the set of Transformer networks with discrete weights. As such, we do not have any prior reason to expect that the discrete programs will be more robust. On the contrary, our primary goal is interpretability, and we expect that Transformer Programs might sacrifice accuracy for this goal. Nevertheless, we show that Transformer Programs can achieve competitive accuracies on synthetic RASP tasks and real-world NLP tasks. We also highlight some areas where our method struggles to learn robust programs, due to the challenges of discrete optimization. Replacing large transformers with drop-in interpretable counterparts is an ambitious project, and we see this work as a first step. Addressing these optimization challenges is an avenue for future work.
>
> > Lack of Reusability
>
> Our experiments show that the Transformer Programs do in fact learn to compose operations, computing primitive values at lower layers and using these results at higher layers to compute higher order patterns. For example: the simple in-context-learning program learns an  induction-head pattern  (section 4.1), and the double-histogram program (appendix Figure 9) learns to compute the first-order histogram in the first layer, and consume this value in the second layer. In Appendix C.2 (Figure 10), we show that higher-layer attention heads often read information from intermediate-layer attention and MLP layers, showing that the programs do indeed learn to reuse previously computed values.
>
> > Results on longer sequence tasks
>
> To clarify, this line is referring to models that are trained on longer sequences, not to OOD generalization. As noted above, our work is not a model distillation approach, and we don’t expect other results about distillation to apply here. The reason the results degrade on longer sequences is related to optimization challenges: our programs tend to find solutions that use relatively shallow features of the input, rather than the most robust, parsimonious solution. As discussed above, addressing these optimization challenges will require significant research advances, which we are leaving for future work.
>
> > Learned solutions vs. human-written solutions
>
> We did not expect in advance that the Transformer Programs would learn the same solutions as the human-written programs for RASP, which represent just one possible way to solve the problems. In fact, in the RASP paper (Table 1), they find that even standard Transformers do not always learn the same solution as the human-written program. As noted above, many of the programs described in the paper learn to reuse previously computed values. However, there could be other inductive biases that make some solutions more difficult for our model to learn–for example, it might be harder for our model to learn arbitrary feed-forward operations.
>
> > What is a variable here?
>
> We use the word variable in the same sense as a variable in a computer program: a named container that takes on a particular value when the program is executed. In the example in section 3.1, the program has four variables: _tokens_, _positions_, _attention 1 outputs_, and _attention 2 outputs_. We follow the design of the Tracr compiler and use a one-hot encoding for categorical variables, meaning the embedding size is equal to the number of variables times the number of possible values they can take on. We will define this more precisely in our final draft.
>
> > requiring each query token attend to a single key token
>
> In this paper, we support two kinds of attention: categorical attention, where each query token attends to a single key token, and numerical attention, where each query token can attend (uniformly) to any number of key tokens. This is a limiting constraint, but we adopt it from RASP/Tracr, and it is important for ensuring that the resulting programs are discrete and easy to interpret. Moreover, existing work has shown that these attention patterns are empirically common in Transformer LMs [4], and, theoretically, are expressive enough to model a large class of formal languages [5].
>
> [4] Merrill et al., 2021. Effects of Parameter Norm Growth During Transformer Training: Inductive Bias from Gradient Descent.
>
> [5] Merrill et al., 2022. Saturated transformers are constant-depth threshold circuits..
>
> > Related work
>
> Thank you for the pointer! We will add a discussion and related references in our final draft.

---

> > ### Comment · Reviewer_TaJd · 2023-08-13
> >
> > Thank you for the clarifications and apologies for the model distillation misunderstanding. After re-reading the paper, I'm raising my score to a weak accept.

---

### Official Review · Reviewer_HW9X · 2023-07-06

**Soundness:** 4 excellent
**Presentation:** 3 good
**Contribution:** 3 good
**Rating:** 7
**Confidence:** 3

**Summary:**

There has been prior work on compiling programs written in a domain-specific language (RASP) into a transformer that emulates the function. In this paper, the authors target the opposite direction of training transformers and generating RASP programs that are faithful to the model's computation. The authors do this by proposing an architecture that serves as a discretized version of a transformer, allowing translation to a discrete, interpretable python program after training. They find that for a variety of tasks, it is possible to 1) learn a solution in this architecture and 2) compile it into RASP.

**Strengths:**

This work demonstrates a novel architecture that allows for direct translation to code. The paper does a great job of demonstrating their method for synthetic tasks. Though prior work has noticed sparsity improved interpretability, this work leverages it for full programmatic translation. I hope this can inspire future work in interpretable neural networks.

**Weaknesses:**

1. Learning solutions to the RASP tasks makes it easier for the model to learn interpretable solutions. The paper would be stronger if it targeted other tasks where the compilation is more surprising.
2. How well can a discrete transformer solve problems relative to the original transformer? I imagine the discretized model has a lower representation capacity or worse optimization. I think it's important to experimentally/theoretically analyze what is lost as a result of this architectural choice, considering it's not a standard transformer in many key aspects.


**Questions:**

No questions beyond the one in the weaknesses/limitations.

**Limitations:**

Are there tasks where you tested this method and either the code was uninterpretable or the discretized transformer couldn't achieve high accuracy, unlike the standard transformer? If not, are there any tasks where you wouldn't expect this to work?

Other than this, the authors have adequately addressed limitations.

---

> ### Author Rebuttal · Authors · 2023-08-09
>
> Thank you for your review! Please also see the GR for our comments about whether the solutions are always interpretable, and about the comparison between Transformer Programs and standard Transformers.
>
> > Learning solutions to the RASP tasks makes it easier for the model to learn interpretable solutions. The paper would be stronger if it targeted other tasks where the compilation is more surprising.
>
> In addition to the RASP tasks, we trained models on two non-synthetic NLP tasks: named entity recognition (Section 4.3) and text classification (Appendix C.3). In the paper, we show examples from the learned programs to illustrate how they allow us to interpret the solution. For example, in Figure 7, we identify some of the features that the program uses to distinguish location entities from organization entities. These include attention heads that copy lexical information from the preceding and subsequent tokens–e.g., a token is more likely to be the beginning of a location entity if the following word is a word like “Germany”, “Italy”, or “Netherlands.” On a question classification dataset (Appendix Fig. 11), we can see that the model focuses on tokens at the beginning of the input–for example, predicting that the input is a numerical question if it starts with a question word like “when”, or relevant nouns like “year”, “time”, and “date.”
>
> We chose to experiment on the RASP tasks because these reflect a variety of algorithmic subproblems. We would also note that, even though these tasks can be solved with concise, interpretable programs, it is still non-trivial to automatically learn these solutions given only input-output pairs.
>
> > How well can a discrete transformer solve problems relative to the original transformer?
>
> The paper and appendix include experimental results comparing the original transformer and our discrete transformers on the NLP tasks: Appendix C.4 (Table 6) shoes that the Transformer Programs are are either competitive with or slightly worse than the standard transformer on four  text classification tasks, and Section 4.3 (Table 2 in the main paper) shows that the Transformer Program actually slightly outperforms a standard transformer in our NER experiments.
>
> On the other hand, the Transformer Programs under-perform a standard transformer on some RASP tasks, and on longer inputs. We did not include RASP results with a standard transformer, but Weiss et al. [1] report that standard transformers get > 99% accuracy on tasks with a sequence length of up to 100 tokens, while our Transformer Programs perform worse on longer sequences (Appendix C.1, Table 3). (We will include RASP results with a standard transformer in our next draft.) This is an optimization issue rather than an expressivity issue: we can write Transformer Programs that achieve perfect accuracy at arbitrary sequence lengths, but our optimization procedure fails to find these programs in practice. We hope to address these optimization challenges in future work. (In general, Transformer Programs can be as expressive as RASP models, with the caveat that we have restricted the size of the feed-forward layers; like RASP models, the discrete transformers are less expressive than standard transformers due to the limitations of hard attention [2].)
> We will highlight this discussion in greater detail in our final draft.
>
> [1] Weiss et al., 2021. Thinking Like Transformers.
>
> [2] Hao et al., 2022. Formal language recognition by hard attention transformers: Perspectives from circuit complexity.
>
> > Are there tasks where the code was uninterpretable or the discretized transformer couldn't achieve high accuracy?
>
> As noted above, the discretized model performs worse than a standard transformer on RASP tasks with longer sequences, and performs slightly worse on some text classification datasets. The Transformer Programs also perform relatively poorly on the Most-Freq task (Table 1), which requires the model to count, sort, and identify unique elements. In general, we might expect Transformer Programs to perform worse on tasks like Most-Freq that require the composition of a series of higher-order functions, because of optimization challenges (i.e. local minima).
>
> Regarding tasks where the code was difficult to understand: In Section 4.2, we highlight some subsets of the _sort_ program with non-obvious interpretations. For example, Figure 6 illustrates a circuit of two attention heads that propagates early-position tokens to later positions and late-position tokens to early positions by first copying them to the beginning- and end-of-sequence tokens. In general, for larger-scale tasks, it might become more difficult to understand the code at the level of individual attention heads. Some interesting directions for future work are  to develop tools for identifying higher-level abstractions in large programs, and for imposing an inductive bias in favor of sparser programs. Please also see our comment in the GR about whether the learned programs are interpretable.

---

> > ### Comment · Reviewer_HW9X · 2023-08-11
> >
> > Thank you for your responses, especially for pointing to the original/discrete transformer accuracy comparisons I missed! In light of this, I have increased my score from a 6 to a 7.

---

### Official Review · Reviewer_vQnY · 2023-07-06

**Soundness:** 4 excellent
**Presentation:** 4 excellent
**Contribution:** 3 good
**Rating:** 7
**Confidence:** 5

**Summary:**

This paper presents an approach to constraint the parameter space of Transformers so that when trained, the resulting weights can be directly translated to a human-readable program, e.g. in Python. To do so, attention is modified so that the internal model representation is, in practice, composed of a set of distinct variables that the attention processes read/write to. The base approach focuses in attention-only models (no MLP layers), but the authors propose extensions to support MLP layers.

- page 5: line 165-166: "For row in the predicate matrix" -> there seems to be some word missing around "row".


**Strengths:**

I found the idea of constraining the parameter space extremely interesting. While the types of programs that can currently be learned seem a bit restricted (e.g., as the number of variables grows, the model/program could potentially grow very large), I think this is a very promising direction, and a greag foundation to build on.


**Weaknesses:**

While it seems clear that programs for simple tasks can be interpreted, it is not clear that many tasks (specially vision/natural language tasks) actually even afford interpretable algorithms. So, perhaps future work needs to focus on more complex primitives for the programming language supported by the models that would simplify the programs, or enable higher-level constructs that are interpretable.


**Questions:**

- Section 3.2: the explanation for how to extract the part of the program that corresponds to the attention layers is clear. But how about the MLP layers? How do you infer what the MLP layers have learned to do to conver that into a program? [edit: this is explained immediately below, but it's still unclear; so, for EACH MLP in EACH LAYER of the model, you would have to extract a look-up table that will be added to the Python program?]
- (more a comment than a question): concerning MLPs. I wonder if it would be possible to define a library of commmon operations (e.g. addition/multiplication/etc., or whatever makes sense with other categorical values), and allow non-learnable, pre-defined MLP weights to be used via some gating (e.g., the model choosing to use one of these built-in methods, or a custom-learned weight matrix). to increase the chance of having an interpretable function in the resulting model, rather than having one look-up table per MLP layer.


**Limitations:**

See weaknesses above.

---

> ### Author Rebuttal · Authors · 2023-08-09
>
> Thank you for the comments!
>
> > Higher-level constructs for more complex tasks
>
> Thank you for the great suggestions! We agree that future work should explore extending the framework to include more complex primitives, or perhaps automatically identifying higher-level programming abstractions in learned programs (e.g. as in [1]). We will mention this direction in our updated draft.
>
> Please also see the GR for our more general comment about whether complex tasks admit interpretable algorithms.
>
> [1] Ellis et al., 2023. DreamCoder: Growing generalizable, interpretable knowledge with wake–sleep Bayesian program learning.
>
> > Converting MLP layers into a program
>
> Thank you for pointing this out – this is correct. For each MLP, for each layer of the Transformer, we extract a lookup table that will be added to the Python program. (However, within a single Transformer layer, each MLP module is converted into a single lookup table, regardless of the number of hidden layers within the MLP itself.) The size of the lookup table is determined by the number and cardinality of the input variables, although in practice we can compress the lookup tables in situations where the model does not use all of the output values. Some example MLP functions are presented in the appendix in Figures 8 and 9. We will update Section 3.2 to explain this more clearly.
>
> > Defining a library of common operations for MLPs
>
> Thank you for the suggestion! This is a great idea and it would be simple to incorporate this type of module into this framework. We chose to treat MLPs as lookup tables following Tracr, but we agree the resulting lookup tables are not necessarily easy to interpret, and incorporating common operations like addition would be a good alternative. We will mention this suggestion in our final draft as a possibility for future work.

---

### Official Review · Reviewer_Kcxc · 2023-07-09

**Soundness:** 3 good
**Presentation:** 3 good
**Contribution:** 3 good
**Rating:** 7
**Confidence:** 3

**Summary:**

This paper proposes a new technique for training Transformer models while ensuring that the learned weights obey certain constraints, designed to facilitate mechanistic interpretability. In particular, the technique works by (1) freezing certain weight matrices throughout training (e.g., to always write the output of an attention layer to a previously unused linear subspace of the residual stream) and (2) introducing Gumbel-softmax sampling at various steps of the computation, and annealing the temperature down during training to ensure that at convergence, those computations perform discrete operations. The resulting Transformers can be automatically converted into symbolic programs that read, write, and perform lookup-table-style computation with a fixed number of categorical and integer variables. The authors train (quite small) Transformers in this way, to solve several tasks, including named entity recognition and sentiment classification on short sequences. The authors also provide some analysis of the programs extracted from these Transformers.

**Strengths:**

This is a neat paper that advances two exciting lines of research:

- RASP and Tracr been helpful as intuition pumps for how trained Transformers *could* work, but it has not been clear whether the (correct) RASP algorithms researchers manually developed could be discovered automatically by gradient descent, and if not, what different properties these learned Transformer algorithms would have. This paper proposes a method for learning Transformer programs that could be used to start to answer these questions.
- A key challenge in mechanistic interpretability research (e.g. on Transformer circuits) is that Transformers trained by (unconstrained) gradient descent appear to collapse multiple conceptually distinct concepts via superposition. This paper proposes training methods that may help to mitigate superposition by forcing the network to read, write, and manipulate individual ‘variables’ that live in orthogonal subspaces of the residual stream.

Although terse in some places (perhaps due to length constraints), the methodology is for the most part clearly described, and the limitations of the approach seem sensible for a first attempt at this kind of constrained training. Another strength is that the framework seems to be extensible—the examples in the paper paint a clear enough picture of how to modularly extend the target symbolic programming language with new features (and then how to extend the neural net architecture and training procedure accordingly).

**Weaknesses:**

- Although the motivation is interpretability, I was not convinced that the learned programs in the experiments were particularly interpretable. It would have been nice to see more detailed analysis of the programs. Outside the toy in-context learning example, are there any interpretable algorithms that are learned? Can examples on which the learned programs fail be traced back to particular ‘bugs’ in the symbolic programs? Can the symbolic programs be used to engineer adversarial examples on which the Transformer will fail? Do ordinary transformers trained without constraints also fail on these adversarial examples (indicating that they might be using similar flawed algorithms)? How stable are the learned programs across different training runs?

- It is unclear how scalable the presented technique is, to more complex problems, longer sequences, or bigger networks. I also would have appreciated more discussion / investigation of how easy or tricky the optimization was, and why it appears to work more poorly for larger Transformers (e.g., that handle larger sequence lengths). What are the key challenges in scaling up? Do the Gumbel gradient estimates become too noisy, e.g.? Or are there other main challenges?

**Questions:**

- At the end of training, your Gumbel-Softmax's have been annealed to Categorical distributions, but these distributions are not necessarily *peaked* or low-entropy categoricals. Therefore, your extracted program (which takes the argmax of the categorical) may behave quite differently from your trained network (which randomly samples the categorical). Have you investigated (a) the extent to which the learned categoricals are peaked at a single mode, and (b) the difference in task performance / accuracy between the final learned Transformer and the extracted program?
- Why limit the predicate matrices to contain only one ‘1’ per row? Is it just because this constraint is more amenable to the Gumbel-softmax training procedure?
- In the description (Appendix A.1) of the categorical attention mechanism, you appear to use an argmax operation that is not differentiable. During training do you actually use Gumbel-softmax? If so, how are the resulting samples used to compute the output of the attention layer? (More generally, it would help if the paper more clearly laid out exactly where in the architecture all the Gumbel-softmax samples are used. Perhaps algorithm boxes in the style of https://arxiv.org/pdf/2207.09238.pdf could be provided for each component of the neural architectures you build?)
- For the in-context learning experiment, did you try (1) training an ordinary Transformer of the same size on the task? and/or (2) training a larger (i.e., overparameterized) Transformer program on the task? I’d be curious about the results of either experiment, if you did run them. Under the ‘lottery ticket hypothesis,’ overparameterization is an important part of how algorithms are learned by gradient descent — it would be interesting to see what this looks like in a Transformer program. E.g., in a larger model, do we end up with a small interpretable subprogram that correctly completes the task?
- Can you clarify how the word embeddings are trained? What does it mean that the 100-d GloVe embeddings are “randomly projected” to e.g. 32*4=128-d embeddings? Do you use a randomly generated dense 100x128 matrix? Then how are the embeddings trained? Is there a Gumbel sample for each of the categorical variables?
- Why not include MLP layers in the NLP tasks?
- For the RASP tasks that were not successfully solved by learned Transformer programs, do you believe this is because (1) good solutions exist, but are not found by gradient descent, or (2) your subset of RASP is too limited and the tasks cannot be solved with this subset? If (2), what are the prospects for handling a broader subset of RASP?

**Limitations:**

Limitations are adequately discussed, but I do think it could be useful to say a bit more about the challenge of scaling this approach -- what are the key obstacles & prospects for addressing them?

---

> ### Author Rebuttal · Authors · 2023-08-09
>
> Thank you for your review! Please see our comments in the GR about whether the programs are interpretable, and the main challenges to scaling up this approach.
>
> > More analysis of the programs
>
> Thank you for the great suggestions! Regarding debugging: We do not necessarily expect that standard transformers will make the same errors, but we agree it would be interesting to measure the extent to which (a) whether standard Transformers and Transformer Programs learn similar solutions, and (b) the similarity between Transformer Programs with different random seeds, and we will add this analysis to our next draft.
>
> > Are the categorical distributions peaked?
>
> We did not investigate these questions in detail but we will run this analysis and add it to our final draft. Anecdotally, we found that the accuracy of the argmax model closely approximates the final training accuracy.
>
> > Why limit the predicate matrices to contain only one ‘1’ per row?
>
> This was a design choice that we made because it leads to programs that are (qualitatively) easier to understand. It is also possible to allow the predicate matrix to have multiple 1’s per row, and using a Gumbel-sigmoid function. In our preliminary experiments, we trained models with this form of predicate matrix and found they obtained similar classification performance but were qualitatively more difficult to understand. We omitted this discussion for lack of space but will add it to the appendix in our final draft.
>
> > Argmax attention / location of Gumbel-softmax operations
>
> Thank you for pointing this out. We use the Gumbel-softmax during training, in the place of hard attention (this is mentioned in section 3.2 but not in the appendix–we will correct this in the next draft). Using the notation from Section 2, the output of the attention layer is defined as $AxW_V$, where A is an attention matrix with each row adding up to one, and S is a matrix of attention scores. With argmax attention, $A_i = \text{One-hot}(\text{arg}\max_j S_{i, j})$. During training, we define $A_i = \text{One-hot}(\text{Gumbel-softmax}(\log (S_i + \epsilon)))$ for some small $\epsilon$. We will explain this more clearly in our next draft, and include a table laying out the different components of the model.
>
> > Additional in-context learning experiments
>
> We did try (2), training larger Transformer programs. We found that over-parameterized models are more likely to learn the correct solution (as measured by the percentage of training runs that achieve perfect test accuracy). For example, if we train a model with two heads per layer, we are more likely to find that one pair of heads implements the induction head, perhaps analogously to the lottery ticket hypothesis. One interesting question for future work is whether we can identify the interpretable subprogram automatically, for example by adapting pruning methods. We will add this discussion to our updated draft.
>
> > Can you clarify how the word embeddings are trained?
>
> Before training, we initialize the word embeddings to $E = E_G  \Pi$, where $E_G \in \mathbb{R}^{|\mathcal{V|} \times 100}$ denotes the GloVe embedding matrix, and $\Pi \in \mathbb{R}^{100 \times 128}$ is a matrix with each entry sampled independently from $\mathcal{N}(0, 1/128)$. This embedding matrix is then split into four $|\mathcal{V}| \times 32$ matrices (one for each variable), and we sample categorical word embeddings by taking a Gumbel sample for each row of each of the four matrices. We will explain these details more clearly in our updated draft.
>
> > Why not include MLP layers in the NLP tasks?
>
> We used one MLP layer in all of the NLP tasks, but this is not clearly reflected in the paper–thank you for pointing this out. We will update the method description to be more clear.
>
> > Explaining the RASP tasks that were not successfully solved
>
> We believe that the main issue is (1): good solutions exist but are not found by our optimization procedure. This is because it is possible to manually construct Transformer Programs that solve RASP tasks perfectly, so the performance limitations must be attributed to optimization challenges. One caveat is the “less than” predicate, which appears in RASP programs for  tasks such as sorting. This predicate is difficult to express using our version of categorical attention, where each query value attends to a single key value, but could be expressed using the one-to-many version of attention discussed above. It is straightforward to extend our approach to include this kind of attention. In our preliminary experiments, we found that, even using one-to-many attention, the  Transformer Programs did not learn the “less-than” predicate. We will add these experiments to our updated draft.

---

> > ### Comment · Reviewer_Kcxc · 2023-08-21
> >
> > Thank you for your response!
> >
> > A couple follow-ups:
> >
> > > This is because it is possible to manually construct Transformer Programs that solve RASP tasks perfectly, so the performance limitations must be attributed to optimization challenges. One caveat is the “less than” predicate...
> >
> > The reason I asked is because it does seem your target language is less general than RASP. One example is predicates like "less than" (which feels quite significant). But it was unclear to me if this is really the only limitation in comparison to general RASP -- clarifying this in the final draft would be appreciated.
> >
> > > During training, we define...
> > Is this differentiable? How do you propagate derivatives through the one-hot, or do you not need to?
> >
> > > We did not investigate these questions in detail but we will run this analysis and add it to our final draft.
> > > we agree it would be interesting to measure the extent to which (a) whether standard Transformers and Transformer Programs learn similar solutions, and (b) the similarity between Transformer Programs with different random seeds, and we will add this analysis to our next draft.
> > Thanks, I look forward to reading these analyses.
> >
> > Thanks also for your general response discussing scaling challenges. Overall, after the author response, I am slightly more lukewarm on the current results, but believe this work represents a good first step toward learning Transformers that can be represented as symbolic programs. I still support publication of the manuscript & am excited to see how the research community builds on these ideas / results in the future!

---

### Author Rebuttal · Authors · 2023-08-09

We thank the reviewers for their thoughtful comments! The reviewers have made a number of great suggestions that we will incorporate into our next draft, including clarifying some details about the method; providing more discussion about optimization challenges; and including some additional analysis of the learned programs. We respond to each of the reviewers in detail, and to some common issues below.

> Are the programs interpretable?

Some of the reviewers questioned whether the learned programs are actually interpretable (Kcxc, HW9X), or whether more complex tasks even admit interpretable solutions (vQnY). We provided a number of examples in Section 4 and Appendix C illustrating interpretable subsets of programs for different tasks, including sorting (Figure 5); NER (Figure 7); double histogram (Appendix Figure 8);  Dyck-2 (Appendix Figure 9), and a classification task (Appendix Figure 11). For more complex tasks, the program can be thought of as a collection of individually interpretable feature functions, rather than a single interpretable algorithm. For example, a circuit in the Dyck-2 program checks for invalid bigrams using an attention layer and an MLP. In the NER program (Fig. 7), the model is more likely to predict that a token is the beginning of a location entity if the following word is a word like “Germany”, “Italy”, or “Netherlands.”

On the other hand, we acknowledge that the learned programs can still be complicated and non-intuitive. (For example, Figure 6 illustrates a subset of a _sort_ program whose interpretation is not immediately obvious.) We would emphasize that this method always provides a baseline level of interpretability: even when the program is complicated, we can trace how information flows between different positions and model components, and we can inspect the program using off-the-shelf code analysis tools, like debuggers. Nonetheless, we agree that the programs can still be difficult to interpret, and we think a good avenue for future work is to explore methods for automatically analyzing the resulting programs, and for imposing an inductive bias in favor of more interpretable programs.

> Scaling, expressivity, and optimization challenges

Several of the reviewers asked if we could provide more details about how the Transformer Programs compare to standard Transformers (HW9X, TaJd); provide more insight into the cases where the Transformer Programs under-perform standard Transformers (Kcxc, HW9X, TaJd); and discuss the key challenges to scaling up this approach. We provided relevant analysis in the paper by experimenting on RASP tasks with longer sequences (Appendix C.1) and on NLP tasks (Section 4.3 and Appendix 3.3), but we add some additional discussion here. In particular, Reviewers HW9X and Kcxc ask whether the main limitation of this method is (a) limited expressivity or (b) optimization challenges.

We think that the main scalability challenges are optimization challenges. In particular, Transformer Programs are expressive enough to represent robust solutions to the majority of RASP tasks, but in practice they do not learn these solutions. This is reflected in the results in Table 3 (in the appendix), where we learned programs for longer sequences. For example, we can construct a Transformer Program that achieves perfect accuracy on the _reverse_ task, by first calculating the sequence length, then calculating the target position indices using a feed-forward layer (_target_index_ = _length_ - _position_ + 1), and then copying the token from the target index. In ablation experiments, we found that Transformer Programs could learn to calculate these intermediate values if they were trained directly, but failed to learn this decomposition when trained end-to-end on the _reverse_ task. We think that the underlying optimization issue is that it is more difficult for these discrete models to escape from local minima. (We will add this discussion to our next draft.)

Addressing these challenges will require more research on discrete optimization. While our approach might not be a drop-in replacement for regular Transformers just yet, we think that our work represents a significant first step towards this goal, and can help lay the groundwork for future research.

---

### Decision · Program_Chairs · 2023-09-21

**Decision:**

Accept (oral)

**Comment:**

This paper introduces a novel method to learn transformer programs, which are executable programs that can be interpreted by a transformer model. The reviewers recognized that the paper presents an interesting and original idea that has potential applications in natural language processing and program synthesis. They also pointed out some issues such as the interpretability and scalability of the generated programs, the optimization challenges, and performance comparing to the baseline Transformer models. The authors provided a rebuttal that addressed most of the issues raised by the reviewers and promised to improve their work in the final version. Based on the reviews, the rebuttal, and the discussion, I recommend an ACCEPT to this paper. I believe this work will inspire further research and applications in this direction and suggest that the authors address the remaining concerns of the reviewers and provide more details and comparisons in the final version.